



# InSAR-derived seasonal subsidence rates reflect spatial soil moisture patterns in Arctic lowland permafrost regions

Barbara Widhalm[1,2], Annett Bartsch[1,2], Tazio Strozzi[3], Nina Jones[3], Artem Khomutov[4], Elena Babkina[4], Marina Leibman[4], Rustam Khairullin[1], Mathias Göckede[5], Helena Bergstedt[1,2], Clemens von Baeckmann[1,2], and Xaver Muri[1,2]

[1]b.geos, Industriestrasse 1, 2100 Korneuburg, Austria
[2]Austrian Polar Research Institute, c/o Universität Wien, Austria
[3]Gamma Remote Sensing, Gümligen, Switzerland
[4]Earth Cryosphere Institute, Tyumen Scientific Centre SB RAS, Tyumen, Russia
[5]Max Planck Institute for Biogeochemistry, Jena, Germany

**Correspondence:** Barbara Widhalm (barbara.widhalm@bgeos.com)

**Abstract.** The identification of spatial soil moisture patterns is of high importance for various applications in high latitude permafrost regions, but challenging with common remote sensing approaches due to high landscape heterogeneity. Seasonal
thawing and freezing of near-surface soil lead to subsidence-heave cycles in the presence of ground ice, which can exhibit magnitudes of several centimeters. Our investigations document higher Sentinel-1 InSAR seasonal subsidence rates for locations with higher near-surface soil moisture compared to dryer ones. Based on this, we demonstrate that the relationship of thawing degree days - a measure of seasonal heating - and subsidence signals can be interpreted to assess spatial variations of near-surface soil moisture. A range of challenges, however, need to be addressed. We discuss the implications of using
different sources of temperature data for deriving thawing degree days on the results. Atmospheric effects must be considered, as simple spatial filtering can suppress large-scale permafrost-related subsidence signals and lead to the underestimation of displacement values, making GACOS-corrected results preferable for the tested sites. Seasonal subsidence rate retrieval which considers these aspects provides a valuable tool for distinguishing between wet and dry landscape features, which is relevant for permafrost degradation monitoring in Arctic lowland permafrost regions. Spatial resolution constraints, however, remain
for smaller typical permafrost features which drive wet versus dry conditions such as high and low centred polygons.

## 1  Introduction

Soil moisture information with high spatial resolution is required for numerous applications in Arctic regions. The saturation of soils determines aerobic or anaerobic conditions and consequently carbon or methane release. Therefore soil wetness representations facilitate the upscaling of fluxes and consequently the determination of greenhouse gas composition (Bartsch
et al., 2023; Miner et al., 2022). Additionally, moisture conditions affect soil thermal properties. Wetter soils transfer heat more effectively and rapidly due to higher thermal conductivity and diffusivity (Farouki, 1981). Higher thermal inertia further leads



to a quicker response to external temperature changes in wetter soils (Campbell and Norman, 1998). Moreover, wet soils can store more heat for a given temperature change because of higher heat capacity (Campbell and Norman, 1998). Thus, soil moisture data is of high importance in Arctic permafrost regions, especially for permafrost and climate modelling (Subin et al.,
2013; Göckede et al., 2017; Zwieback et al., 2019).

Landcover heterogeneity in permafrost lowlands is comparably high with complex spatial patterns of wet and dry soils (Bartsch et al., 2023; Treat et al., 2024). Permafrost related processes frequently lead to changes in landsurface hydrology. This includes drainage and formation of lakes (Nitze et al., 2017; Jones et al., 2011) and ice-wedge degradation, which can be observed on sub-decadal timescales (Liljedahl et al., 2016). Polygonal features are associated with ice wedges. Polygons are
a few meters in diameter and can differ in topography (low centred or high centred) leading to specific wet and dry patterns which change over time (Liljedahl et al., 2016). In general, a change in surface wetness over several years can be associated with permafrost change.

Surface wetness monitoring can be addressed with satellite data, commonly based on data acquired in the microwave domain but also using optical data (Table 1). Thermal observations have been also shown to be of value using the principle of
thermal inertia, but their applicability is limited due to frequent cloud cover and the perturbation of meteorological conditions and vegetation (Zhang and Zhou, 2016). Microwave retrievals make use of the high dielectric permittivity of liquid water in the microwave domain as compared to other soil materials (Barrett et al., 2009). A clear advantage is the ability to penetrate cloud cover, providing the potential for good temporal sampling. For global scale products, near-surface soil moisture is therefore derived from microwave sensors. However, the used methods which are based on backscatter intensity (scatterometer or
synthetic aperture radar) or brightness temperature (radiometers) are of limited applicability in Arctic environments(Wrona et al., 2017; Högström et al., 2018; Kim et al., 2023). The presence of surface water within coarse scatterometer footprints may cause deviations. Wind-induced variations in scattering properties generate biases in lake-rich areas (Högström and Bartsch, 2017). Furthermore, short term variations of soil moisture derived from C band radar are influenced by temperature variations of the organic layer (Högström et al., 2018). For passive microwave sensors radiative transfer models are applied utilizing the
measured brightness temperature, while for active sensors various approaches do exist including change detection and modelling methods (Das and Paul, 2015). Global soil moisture products from active and passive microwave systems like from ASCAT (Advanced Scatterometer, C-band, (Bartalis et al., 2007; Wagner et al., 2010)), AMSR2 (Advanced Microwave Scanning Radiometer 2, multi-frequency, (Parinussa et al., 2015; Zhang et al., 2021)), SMOS (Soil Moisture and Ocean Salinity, L-band, (Kerr et al., 2012; Sadri et al., 2020)) and SMAP (Soil Moisture Active Passive, L-band, (Colliander et al., 2017;
Sadri et al., 2020; Entekhabi et al., 2010)) only provide very coarse spatial resolutions (10-50km) and are therefore not suitable for heterogeneous Arctic landscapes. As an alternative, higher but still comparably coarse resolution static data (75-500m) was applied based on ENVISAT ASAR data (Advanced Synthetic Aperture Radar, active C-band sensor, HH-polarization) depicting spatial wetness patterns in tundra regions using winter minimum backscatter values representing surface roughness as a proxy (Widhalm et al., 2015). Although Quad-pol observations from synthetic aperture radar (SAR) backscatter at C-band
have been shown to be promising (Zwieback and Berg, 2019), these data are usually not acquired. Experiments were made with airborne P-band observations (Ye et al., 2021), but such data are so far not available from space. Recently, Treitz et al.





(2024) utilized in situ surface roughness measurements in conjunction with fully polarimetric RADARSAT-2 data to develop a surface roughness model in a localized study. This model, combined with HH-polarized backscatter and local incidence angle data, was subsequently employed to model a time series of volumetric soil moisture.

Interferometric models using SAR data (InSAR) have previously been created to elucidate how changes in dielectric constant, attributed to time-varying soil moisture, affect interferometric phase (De Zan et al., 2014). In recent years attempts have been made to derive soil moisture changes from related closure phases data (e.g. Michaelides and Zebker (2020); Wig et al. (2023); De Zan and Gomba (2018)). However, as this requires a triplet of interferograms, with one spanning over all three acquisition dates, this application is not feasible in areas with rapid loss of the degree of interferometric coherence as can be
the case in permafrost regions, depending on the used frequency. An alternative are wetness indices which can be derived from multispectral data. Usually bands with reflectance in near-infrared and short-wave infrared are used in combination, e.g. for the Normalized Difference Moisture Index (NDMI, (Cheţan et al., 2020)), or the Tasseled Cap Wetness index (TCW, (Frappier et al., 2023)), which uses a transformation of multiple visible, near-infrared and shortwave infrared bands. These indices, however, do not give actual volumetric soil moisture content, but rather serve as proxies for soil wetness. Although spatial
resolution of multispectral data is substantially higher than for global soil moisture products, temporal sampling is limited due to the requirement for cloud-free conditions in frequently cloud-covered Arctic regions (Sudmanns et al., 2020).

   Another index that can serve as a proxy for soil moisture is the Topographic Wetness Index (TWI, (Riihimäki et al., 2021)), which uses topographic information to depict steady-state soil moisture distribution. This index is for example widely used in carbon research (e.g. Mishra and Riley (2012); Obu et al. (2017); Virkkala et al. (2021)). It's applicability as a soil moisture
indicator is, however, limited as it solely depends on topographic information, which is merely one component influencing spatial soil moisture patterns (Riihimäki et al., 2021).

   InSAR has been already used for a range of permafrost monitoring applications across the Arctic (Bartsch et al., 2023). Information derived from InSAR is expected to provide insight into active layer and soil properties (Schaefer et al., 2015; Chen et al., 2023; Li et al., 2023). Local InSAR seasonally aggregated subsidence patterns have been reported to be related to
wetness gradients (e.g. Liu et al. (2010); Strozzi et al. (2018); Bartsch et al. (2019, 2023)). Permafrost regions are characterized by a continuous period of frozen soil conditions. Seasonal phase change occurs, with gradual thaw followed by gradual freeze of the so called 'active layer'. The phase change results in a volume change, depending on the amount of ice/water in the soil. A high subsidence in summer is expected when the water/ice content of the soil is high. Subsidence typically peaks towards the end of the unfrozen period, usually in late August in Arctic permafrost regions. However, the timing of this peak is subject
to variability influenced by factors such as latitude, local climatic conditions, and interannual fluctuations. Consequently, variations in the peak thaw layer thickness's timing might cause it to occur sooner in August or continue into early October. Seasonally aggregated vertical displacement is in the order of a few centimeters, which can be captured with InSAR techniques (e.g. Strozzi et al. (2018)). The magnitude can vary from year to year depending on the warming of the soil or changes in water content through variations in the water budget.

Bartsch et al. (2019) suggested that the temporal evolution of seasonal subsidence reflects differences in soil properties based on vegetation patterns which reflect soil conditions. Scheer et al. (2023) confirmed the linkage to ground ice content



**Table 1.** Summary of remote sensing techniques for near-surface soil moisture estimation (modified after Engman (1991); Moran et al. (2004); Wang and Qu (2009))

| Sensor type | Approach/ observed property | Basic principle | Advantages | Disadvantages | Applicability in permafrost regions |
|---|---|---|---|---|---|
| Optical | reflectance indices and derivatives such as NDVI | albedo | high spatial resolution, broad coverage | cloud coverage, strongly perturbed by vegetation and other hampering factors | limited applicability due to frequent cloud coverage in Arctic regions (Sudmanns et al., 2020) |
| Thermal Infrared | surface temperature | thermal inertia | high spatial resolution, broad coverage | cloud coverage, strongly perturbed by vegetation, influenced by meteorologic conditions | limited applicability due to frequent cloud coverage in Arctic regions (Sudmanns et al., 2020) |
| Passive Microwave | brightness temperature | dielectric properties | high temporal sampling, broad coverage, all weather | low spatial resolution, perturbed by surface roughness and vegetation | limited applicability in heterogeneous Arctic (Wrona et al., 2017), (Kim et al., 2023) |
| Active Microwave | backscatter | dielectric properties | high spatial resolution, all weather | perturbed by surface roughness and vegetation | low reliability in Arctic regions (Högström et al., 2018), (Kim et al., 2023) |
| Active Microwave | backscatter | surface roughness | high spatial resolution, all weather | only in Arctic regions, only static product containing 4 classes | applicable in Arctic tundra regions (Widhalm et al., 2015) |
| Active Microwave | InSAR surface displacement | volume change | high spatial resolution | only in permafrost regions, only one static product per thawing season, atmospheric effects | applicability in Arctic permafrost regions tbd |

although only a limited number of samples were available. Chen et al. (2020) and Chen et al. (2023) suggested the retrieval of equivalent water depth from InSAR subsidence. Seasonally aggregated subsidence may thus hold the potential to serve as proxy for near-surface soil moisture similar to indices based on multispectral data which represent general landsurface wetness and are available at a similar level of detail (including e.g. NDMI from sensors such as Landsat or Sentinel-2).

Temporal resolution is limited to year to year changes when data are seasonally aggregated, but is potentially of interest for permafrost related long-term soil moisture change. Long-term (aggregated over several years) subsidence itself is usually interpreted as a sign for loss of ground ice of the underlying permafrost, as an impact of climate change (e.g. Liu et al. (2015); Wang et al. (2022)). Such an approach does, however, require signal stability, and it necessitates an adequate degree

of coherence in yearly interferograms to establish connections between data from consecutive years, which can be achieved in areas with limited vegetation growth only (Strozzi et al., 2018).



One of the major challenges for InSAR applications in Arctic permafrost regions is the limited availability of regular and spatially continuous SAR acquisitions (Bartsch et al., 2023). A second issue is the available wavelength. Longer L-Band is less sensitive to vegetation changes and is therefore less prone to coherence loss, allowing longer intervals in-between acquisitions.

Shorter wavelengths such as C-band and X-band are usually only applicable in regions with low vegetation. X-band is in general not freely available and time series are limited to small regions. With Sentinel-1 A and B, a freely accessible data set in the C-band range is available, usable for InSAR in Arctic permafrost regions with limited vegetation cover (Sentinel-1A launched in April 2014, Sentinel-1B launched in April 2016 and ended December 2021) (Strozzi et al., 2018). L-band is currently also mostly acquired on demand across the Arctic, however usually only one or two acquisitions are existing for the

unfrozen period, whereas rarely any acquisition matches the timing of the maximum active layer depth. A general challenge is the impact of snowmelt in spring. Signal decorrelation reduces the availability of image pairs at the time when ground thaw is initiated.

A further obstruction for such an application are atmospheric effects and ionospheric activities which disturb the signal (e.g. Muskett (2017)). When atmospherically contaminated interferograms cannot be simply discarded (as, for example, done in Liu

et al. (2010)) owing to coherence restrictions, atmospheric corrections are a necessity. Available atmospheric correction methods can be divided into methods with and without external data (Xiao et al., 2021). External data, such as weather reanalysis or global positioning system soundings, can be utilized to mitigate turbulent atmospheric noise from affected interferograms, when sufficiently available (Dini et al., 2019; Jolivet et al., 2014, 2011; Michaelides et al., 2021). Corrections like stacking or time series analysis, which do not rely on external data but rather on data redundancy of interferogram networks, often do

not capture the complexity of atmospheric effects (Xiao et al., 2021). On the other hand, methods incorporating external data, like ground observations, satellite observations or numerical weather models increase processing efforts (Xiao et al., 2021). The Generic Atmospheric Correction Online Service for InSAR (GACOS) tackles this problem by providing easy to apply corrections which use external information on atmospheric conditions (Iijima et al., 2021). By incorporating weather model data as well as topographic information, Zenith Tropospheric Delay (ZTD) maps are produced and made available in near real

time. The effectiveness of this method has been demonstrated locally (e.g. Murray et al. (2019); Ulma et al. (2021)), in some cases however with varying degrees of success (e.g. Wang et al. (2019)). Particularly capturing the smaller scale turbulent atmospheric phase appears to be lacking, while the mitigation of elevation-dependent and long-wavelength components seems to be feasible (Li et al., 2022). The utility of this approach for permafrost applications has not yet been evaluated so far. This requires testing with in situ data as well as common statistical assessments of phase residuals of the multi-baseline processing.

In situ subsidence data are scarce, but the assessment of phase residuals can be applied independently.

An atmospheric correction method frequently applied in permafrost-related studies is spatial filtering, such as high-pass filtering (e.g. Strozzi et al. (2018); Michaelides et al. (2021); Rouyet et al. (2021)). This approach is also applicable for InSAR datasets of a daisy-chain network, that were processed in series and do not include overlapping interferograms in the time domain. The filtering method relies on the significant disparity in correlation length in the spatial frequency domain of

atmospheric and thaw subsidence induced effects (km- compared to m-scale) (Michaelides et al., 2021).



For seasonal aggregation, the timing of acquisitions in summer in relation to the day of year (DOY) presents another difficulty. The driver of ground thaw, the seasonal warming, can be expressed through the sum of positive degree days/degree days of thaw (DDT). Bartsch et al. (2019) suggest the use DDT to facilitate the comparison of seasonal deformation across different years. Scheer et al. (2023) implemented a DDT dependent methodology in combination with a normalization step.
The approach of DDT requires the availability of temperature data, which is available locally in very few cases only. Therefore, the suitability of reanalyses data needs to be assessed for application across larger regions.

A further challenge is that in situ subsidence measurements as well as soil moisture measurements are scarce across the Arctic primarily due to logistic constraints (Högström et al., 2018; Strozzi et al., 2018; Bartsch et al., 2019). Spatially distributed measurements are necessary in order to capture landscape heterogeneity, limiting soil moisture measurements to the near-
surface and mostly snapshots in time. However, both surface soil moisture and subsidence measurements are available from a long-term monitoring site on central Yamal in Western Siberia and in NW Canada. They can potentially serve as reference sites in addition to campaign data.

In summary, the investigation of InSAR-derived aggregated seasonal subsidence rates as soil moisture indicator necessitates consideration of various retrieval challenges and assessing the applicability across different permafrost landscape types.
The purpose of this study is to investigate the potential of seasonally aggregated InSAR subsidence signals for retrieving a soil moisture indicator index. An interpretation scheme of the relationship between DDT and Sentinel-1-based subsidence for soil moisture categories is derived and its performance is compared to other existing soil moisture approaches using near-surface in situ soil moisture measurements. To ensure reliable results, correcting InSAR values for atmospheric effects is essential. Thus, we assess various methods for correcting these disturbances using in situ subsidence measurements. Regions
with varying landscape histories are selected, including sites with relevant in situ measurements and auxiliary measurement.

## 2   Study regions

Three primary study regions with in situ soil property information were investigated within this research (Inuvik in northwestern Canada, central Yamal in northwestern Siberia, and Chersky in northeastern Siberia, Figure 1 and Table 2). These regions cover permafrost related long-term monitoring sites. Two of these regions with in situ subsidence measurements and spatially
distributed in situ surface soil moisture records were selected for detailed assessment (Inuvik and central Yamal), with one site also offering near-surface as well as borehole temperatures for DDT investigations (central Yamal). Chersky was selected to add to the statistical evaluation of processing results through analyses of the standard deviation of phase residuals. This third region also provides in situ air temperatures and borehole temperatures.

The study regions lie in the zone of continuous permafrost, with Yamal also showing areas of discontinuous permafrost in
the southwest of the processed scene (permafrost extent data source: Obu et al. (2018)). Long-term permafrost monitoring is available in these regions. In northeastern Siberia the North-East Science Station (NESS) was established in 1980 near the city Chersky. The Yamal study region includes the research station Vaskiny Dachi, where permafrost studies have been conducted



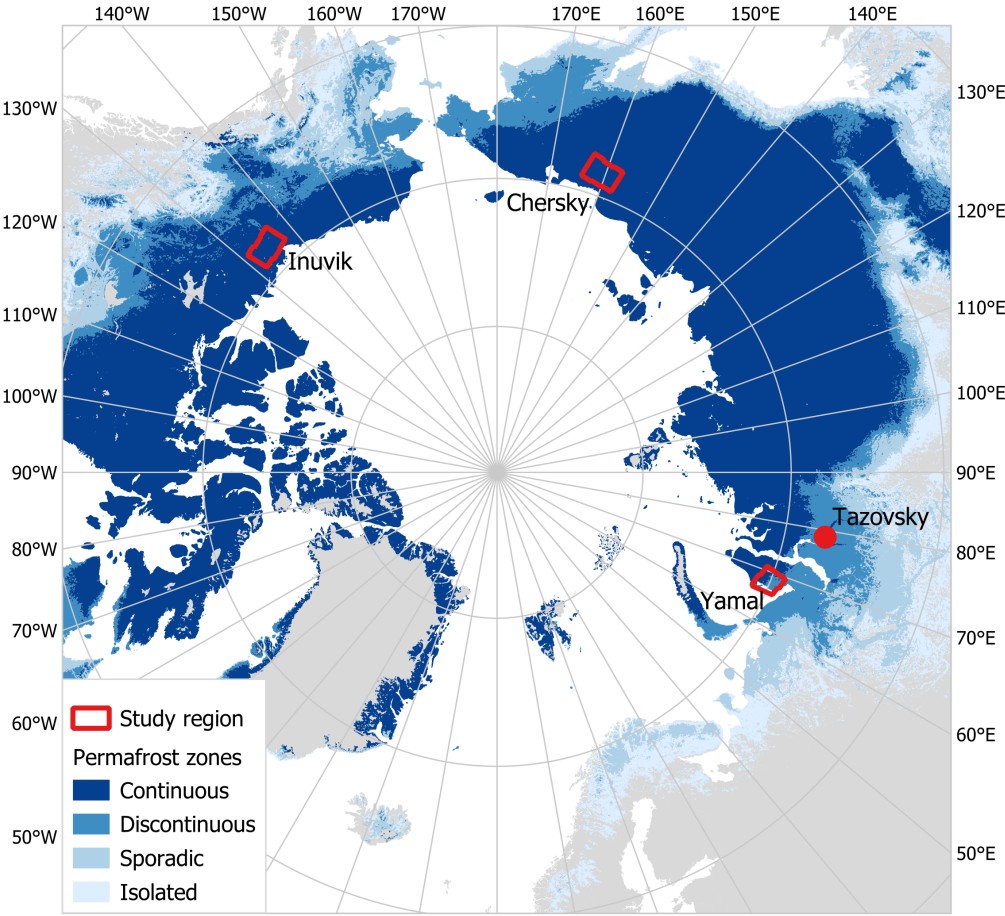

**Figure 1.** Study region locations: Chersky, northeastern Siberia; Inuvik, northwestern Canada; Yamal, northwestern Siberia; Tazovsiky, northwestern Siberia. Background map: Permafrost zones (Obu et al., 2019)

since 1988 (Leibman et al., 2015). The Trail Valley Creek research station (established in 1991; (Pomeroy et al., 1998)) is located within the analysed Inuvik region. Various landscape types are represented within the selected regions.

The lake-rich Khalerchinskaya tundra is located northwest of the lower Kolyma River, in the northwest part of the scene of the Chersky area. This region is underlain by ice-poor sands and dominated by a predominantly waterlogged landscape (Fyodorov-Davydov et al., 2004). The river valley is characterized by alluvial deposits (Shmelev et al., 2017) and features regularly flooded shrub and herbaceous vegetation (GLC). The east and south of the scene is partly underlain by organic- and ice-rich Yedoma deposits (Grosse et al., 2013) featuring needle-leaved and deciduous tree cover as well as herbaceous land 175  cover on mountain regions (GLC).

The Yamal area is characterized by marine, coatal-marine and fluvial-marine lithology (Leibman et al., 2015) where shrub and herbaceous tundra dominates (Khitun and Rebristaya, 1998; Widhalm et al., 2017a; Stolbovoi and McCallum).



The Mackenzie delta area of the Inuvik region is covered by needle-leaved evergreen trees (GLC) and is characterized by alluvial sediments (Geological Survey of Canada, 2014). The west of the scene is dominated by shrub cover and sparsely

herbaceous and shrub covered mountains (GLC). To the west and the east of the delta glacial sediments prevail (Geological Survey of Canada, 2014). East of the delta the land cover is dominated by a mosaic of tree cover and other natural vegetation with areas of shrub and herbaceous cover (GLC).

In order to support the discussion on the applicability of InSAR for permafrost-related features prone to degradation, a fourth region was investigated in an area of discontinuous permafrost near the Tazovsky settlement, about 500km southeast of

the Yamal sites. The characteristics of polygonal tundra, which may vary in moisture regimes, are discussed, highlighting the relevance of InSAR results for characterizing different types of permafrost wetlands. This region is characterized by a relatively flat, slightly dissected surface with a high number of wetlands, lakes and widely spread flat-topped and convex-hummocky peatlands (Babkin et al., 2018). The temperature of permafrost ranges from 0 to –1.0°C, decreasing to –1.5°C in dome-shaped peatlands (Koroleva et al., 2021). Polygonal features are widespread in this area and are usually linked to peatlands (Khomutov

et al., 2022). The features include high- and low-centered polygons. Low-centered polygons have comparably wet soils and/or water-filled ponds inside them. Further landscape features include watery troughs on peatlands, over-saturated wetlands and wetlands without visible polygons.

The active layer (maximum thaw depth; source: ESA Permafrost_cci, year 2019 of Obu et al. (2021)) is largest for the Inuvik study region and the region around Tazovsky with on average 0.96 m and 0.94 m respectively. Yamal shows average active

layer values of 0.84 m and the Chersky area features lowest values with circa 0.64 m.

## 3 Data

### 3.1 Sentinel-1

ESA's Sentinel-1 satellites are operating C-band SAR instruments with a wavelengths of 5.6 cm. For our investigations we used Single Look Complex (SLC) images in Interferometric Wide swath (IW) mode, which provide a ground sampling distance of

2.3 m in range and 13.9 m in azimuth direction and a swath width of 250 km. Sentinel-1A was launched in April 2014 and Sentinel-1B followed in April 2016. The constellation of two satellites offered a possible repeat cycle of 6 days, however due to global acquisition strategies 12 days were more common and was also generally available for most Arctic regions. Regular acquisitions started for most study regions in mid-2016, not always covering the entire thawing season of this year. End of 2021 Sentinel-1B stopped operating resulting in no recent Sentinel-1 data acquisitions for the Siberian study regions.

Supplement Table S 1 indicates the acquisition dates of all Sentinel-1 SAR acquisitions used in this study. Paths and frames of the investigated scenes can be found in Supplement Table S 2. The seasonal study periods were selected from the start of the thawing season, as soon as interferograms showed good coherence, until the onset of freezing (delineated from ERA5 reanalysis data).



## 3.2   In situ data

In-situ data, including subsidence, soil moisture and temperature measurements were available for the three study regions. An overview of the used data is provided in Table 2.

### 3.2.1   Yamal

Yearly in situ subsidence measurements were conducted at the CALM (Circumpolar Active Layer Monitoring) grid near the research station Vaskiny Dachi on central Yamal. For this, L-shaped metal rods are inserted through holes in plates to the base of the active layer, with the hooked end tight on the plate. In winter, frost heave raises the plate, pushing the rod up, where it becomes fixed by freezing. During spring thaw, the plate subsides with the ground surface, while the rod remains fixed. The difference between the plate and the rod's end is measured as ground movement. Surface subsidence measurements are starting in 2016 with 5-6 points. Additional 20 points are available since 2018 and further 4 points since 2021. Measurements were performed yearly end of August or start of September with the exception of 2016 where travel restrictions resulted in measurements not until mid of October. The CALM grid is situated on a sloping plane where dry cryptogram crust dominates the flatter upper part. Grasses and mosses as well as low and high shrubs can also be found in the remaining parts along with patches of wet sedge (Widhalm et al., 2017a).

Soil moisture measurements were conducted at the 121 CALM grid points (Widhalm et al., 2017b) and at 7 transects containing 12 to 34 individual points, as well as at 10 selected additional locations. Moisture values were measured at the top 5 cm with Delta-T Devices HH2 soil moisture sensor. For the soil moisture values at the CALM grid measurements of 3 different dates in August 2015 were averaged. The other points were measured only once, also in August 2015. The moisture conditions in the Yamal Peninsula study region were normal in 2015, with total precipitation values of 185 mm for the months of June through September, compared to median values of 188 mm over a 15-year period (measured at Maresale). Precipitation values during the years of InSAR observations ranged from 120 mm in 2017 to 244 mm in 2021. For the comparison with InSAR results soil moisture values of points which were located within the same Sentinel-1 pixel were averaged. This resulted in 132 samples (68 CALM grid values, consisting of 121 valid samples and 64 pixels in the other categories, containing 146 valid measurements).

Near-surface soil temperatures, for the derivation of DDTs based on in situ data, were measured with DS1921G-F5 Thermochron iButtons between October 2016 and August 2017. Temperatures were recorded at 4 sites at the Yamal study region (locations shown in Figure 2). One point is located at the CALM grid, while the other 3 points are placed at monitoring sites which were established within the Greening of the Arctic (GOA) project of the International Polar Year (IPY) (Walker et al., 2009). Site VD1 is located on a gentle terrace hill-top with clayey soils and sedge and dwarf-shrub, moss tundra. VD2 is on a broad hill-top, characterised by sandy and clayey soils dominated by dwarf-shrub, graminoid, moss tundra. The site VD3 is characterised by dry dwarf-shrub-lichen tundra on sandy soils (Walker et al., 2009).

Additionally borehole temperature data at 50 cm depth of two of the GOA points (VD1 and VD2) were used.





### 3.2.2 Chersky

Borehole data from depths of 4 cm and 8 cm, as well as temperature data from an automatic weather station near the borehole located approximately 15 km south of the city Chersky at the research site Ambolikha, were utilized. This site is located on the floodplain of the Kolyma River and dominated by wet tussock tundra with tussock-forming sedges and cotton grasses on
an organic peat layer overlain by alluvial mineral soils (Göckede et al., 2019).

### 3.2.3 Inuvik

At the Inuvik study site, in situ soil moisture measurements were conducted similarly to those on Yamal, using a Delta-T Devices HH2 soil moisture sensor at the top 5 cm. The measurements were conducted in the region between north of the city Inuvik and south of the Trail Valley Creek research station in July 2023. As this time was characterised by drought conditions,
the maximum of multiple samplings was further used at each measurement point. The measurement points were recorded in transects and irregular point locations. Again, samples within the same InSAR pixel were averaged for comparisons with InSAR results. 78 pixels were used including 91 measuring points.

To investigate the effects of soil moisture change on InSAR results, soil moisture time series at Trail Valley Creek from Boike et al. (2023) were analyzed. Soil moisture data at a depth of 5 cm for the years 2018 - 2022 were used.

In situ subsidence data was available from Anders et al. (2018) for the years 2015 and 2016. Similar to the measurements on Yamal, poles were anchored below the active layer. Measurements were performed twice per year, recording the distance between the top of the pole and a plate on the surface.

### 3.3 Auxiliary data

Air temperature, utilized for deriving DDT values across the entire study regions, was derived from ERA5 reanalysis data.
ERA5 combines model information with observations to produce a globally consistent dataset (Hersbach et al., 2023). We used air temperature at 2 m above the surface in a temporal resolution of 2 hours at 0.25° spatial resolution.

Landsat 8 Level-2 (Bottom of Atmosphere) data (30 m spatial resolution) were acquired for the derivation of NDMI, which utilizes bands in near-infrared and short-wave infrared spectrum to depict changes in water content of leaves. Dates were selected close to the in situ soil moisture sampling dates, specifically on 10.08.2015 for Yamal and on 06.07.2023 for Inuvik.

For the calculation of TWI, the ArcticDEM at 2 m spatial resolution was utilized, which is delineated from optical stereo imagery.

ESA's CCI Soil moisture product (Gruber et al., 2019; Dorigo et al., 2017; Preimesberger et al., 2021) was used for evaluation purposes. This daily, global soil moisture product, at 0.25° spatial resolution, combines various active and passive microwave products. Here, we utilized the passive microwave and combined (active and passive) product, which are provided in volumetric
soil moisture units.

High resolution satellite imagery of Quickbird-2, WorldView-2 and WorldView-3 (Khairullin et al., 2019), topographic surveys (Babkin et al., 2018), and unmanned aerial vehicle (UAV) images were available for the area surrounding the Tazovsky



(a) Chersky

(b) Inuvik

(c) Yamal

**Figure 2.** Sample point and in situ data locations (see also Table 2 for in situ data description overview). (a) Chersky region: sample points for time series plots (without in situ information) and borehole location (Background map data source: Google Satellite). (b) Inuvik region: locations of in situ near-surface soil moisture measurements and in situ subsidence data (Background map data source: ESRI Satellite). (c) Yamal region: locations of in situ near-surface soil temperature and soil moisture measurements (Background map data source: ESRI Satellite. (b) and (c) show only a subset of the processed region, depicting the area where in situ data is available.



**Table 2.** In Situ subsidence, soil moisture and temperature data overview (for locations see Figures 1 and 2)

| Parameter | Region | Site name | Distribution | Sampling type | Dates | Nr. of used samples / sites |
|---|---|---|---|---|---|---|
| subsidence (plates) | Yamal | CALM | irregular point locations | yearly measurements | 2016 - 2021 measured end of summer | 6 - 27 per year |
| subsidence (plates) (Anders et al., 2018) | Inuvik | Trail Valley Creek | irregular point locations | 2 measurements per year | 09.06.2015, 20.08.2015, 18.07.2016, 23.08.2016 | 2 |
| near-surface soil moisture (Delta-T probe) | Yamal | CALM | regular grid | temporally averaged | 19.08.2015, 23.08.2015, 27.08.2015 | 121 |
| near-surface soil moisture (Delta-T probe) | Yamal | other | transects & irregular | single measurement | Aug. 2015 | 146 |
| near-surface soil moisture (Delta-T probe) | Inuvik | | transects & irregular | spatially averaged | Jul. 2023, drought conditions | 91 |
| near-surface soil moisture (Boike et al., 2023) | Inuvik | TVC | fixed point location | regular time series (1h interval) | 2018-2023 | 1 |
| near-surface soil temperature (iButtons) | Yamal | CALM, VD1, VD2, VD3 | spatially distributed fixed point locations | regular time series (4h interval) | 01.01.2017 - 24.08.2017 | 4 |
| temperature, 50 cm depth (borehole) | Yamal | VD1, VD2 | spatially distributed fixed point locations | regular time series (daily interval) | 01.01.2017 - 24.08.2017 | 2 |
| temperature, 4 cm, 8 cm depth (borehole) | Chersky | Ambolikha | fixed point location | regular time series (daily interval) | 2017 | 1 |
| air temperature (AWS) | Chersky | Ambolikha | fixed point location | regular time series (daily means) | 2017 | 1 |





settlement. The UAV images, acquired in 2022, covered an area of about 25 km$^2$. Additionally, high-resolution satellite images available via Google and Esri Satellite maps were used in the proximity.

The Generic Atmospheric Correction Online Service for InSAR (GACOS (Yu et al., 2018)) was developed at Newcastle University and provides high spatial resolution Zenith Tropospheric Delay (ZTD) maps based on numerical weather models. Surface pressure, temperature and specific humidity from High Resolution ECMWF weather model at 0.1° and 6h resolutions is used as input as well as the 90 m resolution SRTM DEM (60°S - 60°N) and ASTER GDEM (at higher latitudes). An iterative tropospheric decomposition model (Yu et al., 2017) is implemented in order to separate the stratified and turbulent components

from the tropospheric delays and produce ZTD maps, which are globally available in near real time.

## 4   Methods

The described method is based on seasonal freezing and thawing of the active layer in permafrost regions. In the presence of ice in the ground, the surface subsides throughout the thawing season. This can be measured using InSAR. Bartsch et al. (2019) demonstrated that the displacement values follow a nearly linear progression with respect to DDT. An overview of the

major processing steps is given in Figure 3. We utilize this relationship in order to link satellite data to in situ near-surface soil moisture, which was collected for calibration and validation purposes. Special attention is payed to interfering atmospheric effects, by testing various correction methods.

    InSAR displacement results and atmospheric correction performance are evaluated by investigating displacement time series at sample point locations across all study regions. A comparison to in situ subsidence data is performed for the Yamal site, and

standard deviations of phase residuals are compared for unfiltered and GACOS-corrected results across all study regions.

    GACOS corrected results and comparable soil moisture indices are evaluated against in situ soil moisture measurements of Yamal and Inuvik. While in situ soil moisture measurements were limited to the top 5 cm, we posit that under typical conditions, they are representative of the entire active layer, which can be measured using InSAR. An empirical relationship between GACOS corrected results and in situ soil moisture of Yamal calibration data (normal moisture conditions) is derived

and its quality compared to the CCI soil moisture product. An overview of the conducted evaluations is provided in Table 3.

### 4.1   InSAR processing

Our InSAR processing sequence largely follows the workflow outlined by Strozzi et al. (2018). This includes the application of precise orbit files, co-registration and computation of interferograms with a multi-looking factor of 5 by 1 pixels. The interferograms were processed in series in a daisy-chain network. Longer time-steps which would result in temporally overlapping

interferograms were often not possible due to decrease in coherence. Temporal baselines were therefore mostly 12 days, in some cases also 24 or even 36 days, when acquisitions were missing or due to apparent coregistration errors or heavy influence of turbulent atmosphere, which lead to the exclusion of isolated acquisitions. Perpendicular baselines were on average 46 m, with a maximum value of 152 m. The next processing steps comprise topographic phase removal, incorporating the Copernicus DEM with 30 m spatial resolution, adaptive phase filtering (Goldstein and Werner, 1998), phase unwrapping (Costantini, 1998),





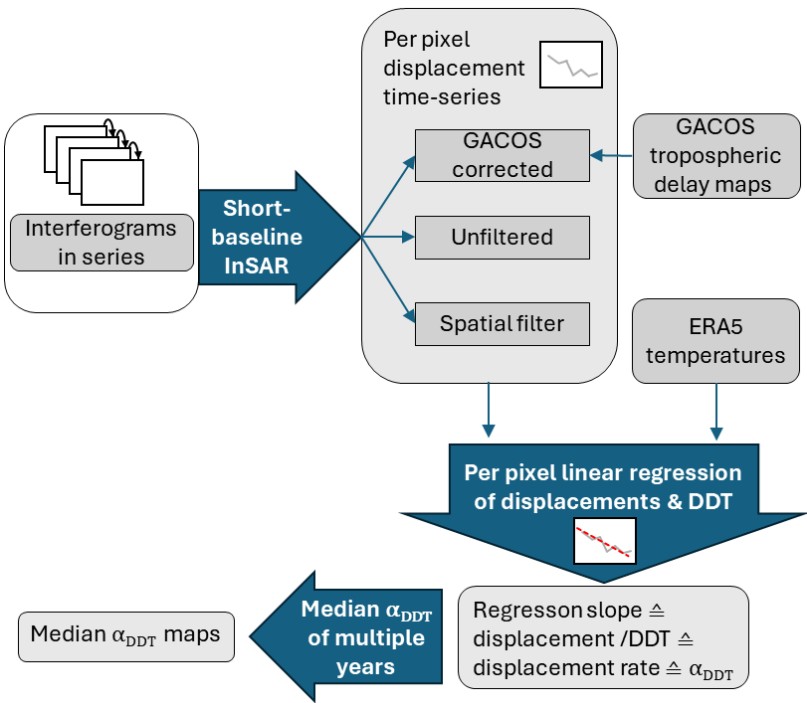

**Figure 3.** Overview of the workflow used for the derivation of median surface displacements in degree day of thaw (DDT) domain ($\alpha_{DDT}$).

**Table 3.** Evaluation and comparison of atmospheric correction and soil moisture indices across various study regions. (* in situ soil moisture measurements represent drought conditions, $^{+}$ permafrost polygonal features spatial distribution)

| Soil moisture indicators assessment | Yamal | Inuvik | Chersky | Tazovsky |
|---|---|---|---|---|
| subsidence rate $\alpha_{DDT}$ (Sentinel-1) | X | X* | | X$^{+}$ |
| Normalized Difference Moisture Index (Landsat) | X | X* | | |
| Topographic Wetness Index (ArcticDEM) | X | X* | | |
| CCI Soil Moisture | X | | | |
| **Atmospheric correction assessment** | | | | |
| displacement time series visualization of sample points | X | X | X | |
| in situ subsidence comparison | X | X | | |
| standard deviations of phase residuals | X | X | X | |
| **DDT retrieval discussion** | | | | |
| In situ ground temperature | X | | X | |
| In situ air temperature | | | X | |
| Reanalyses air temperature | X | | X | |



and calculation of vertical displacements via short-baseline InSAR (Berardino et al., 2002) (assuming that all the displacement is vertical) and terrain-corrected geocoding. The assumption of only vertical displacements holds true for the investigated in situ site locations where only low slopes were being observed and horizontal displacements caused by mass movements, such as solifluction can be ruled out. Areas with slopes of $> 5°$ were subsequently masked to ensure the validity of this assumption. Reference points were selected at or close to airstrips of Inuvik (bedrock outcrop 6 km southeast of airstrip), Chersky and

Bovanenkovo (Yamal region). Areas of low coherence were masked out (average coherence $< 0.8$ and average coherence of filtered interferograms $< 0.5$).

Opting for a daisy-chain network reduces atmospheric effects to the difference between the first and last scenes, which is a an advantage of this processing method. However, this approach also increases noise in the integration, including that related to soil moisture changes. The use of a daisy-chain network makes it challenging to distinguish between surface deformations and

atmospheric effects. These atmospheric artifacts, along with phase delays possibly arising from soil moisture changes (which are considered as a possible limitation and will be discussed later), can introduce noise and inaccuracies into the interferometric phase, leading to errors in the estimated deformation signals. Therefore two different compensation methods for atmospheric effects were tested. First, a spatial filter of the linear-least-squares type, as implemented in the GAMMA software, was applied to the displacement maps. Different filter radii were assessed. Secondly, GACOS corrections were applied on unwrapped

interferograms. Artefacts were encountered in the GACOS products for some study regions (Supplement Figure S 1), stemming from the ASTER DEM for areas north of 60°N used by the provider (Yu et al., 2018). The data provider offers the possibility to send in an alternative DEM for a requested area. However, in this case we opted for correcting the GACOS files by masking out the artefacts and filling in the missing values with the median of a moving window.

## 4.2 DDT and subsidence relationship

The data gap at the beginning of the thawing season, caused by low coherence values due to snow cover on the ground, was accounted for by extrapolating the time series using linear regression (Bartsch et al., 2019). This ensured that every depicted seasonal displacement time series starts with the onset of thaw at DDT = 0. The displacement values calculated for the comparison to in situ data were then offset using the slope of the regression of the displacement time series. Utilising the assumed linear relation between DDT and seasonal surface subsidence, we derived the subsidence rate (displacement per DDT,

hereinafter referred to as $\alpha_{DDT}$) at each pixel (Equation 1).

$$\alpha_{DDT} = \frac{n(\sum_{i=1}^{n} DDT_i d_i) - (\sum_{i=1}^{n} DDT_i)(\sum_{i=1}^{n} d_i)}{n(\sum_{i=1}^{n} DDT_i^2) - (\sum_{i=1}^{n} DDT_i)^2} \tag{1}$$

$d_i$ represents the total displacement between the first acquisition and time-step $i$, and $n$ represents the maximum number of available dates.

Using $\alpha_{DDT}$ allows for a calculation of displacement indices that are independent of acquisition time. However, it should

be noted that subsidence may precede at a faster rate at the non captured beginning of the thawing season, compared to the later season (Schaefer et al., 2015). This can introduce a potential source of error. The DDT was derived from the mean air



temperature at 2 m height of daily averaged ERA5 data with a coarse spatial resolution of 0.25°, which might introduce a source of uncertainty compared to the higher resolution of Sentinel-1 data. We calculated $\alpha_{DDT}$ for each thawing season. In order to derive general displacement patterns we calculated the median $\alpha_{DDT}$ values of all processed years. Using the median

value aids mitigating the effect of remaining atmospheric disturbances within some of the processed years. The median $\alpha_{DDT}$ was further used to investigate the relationship of InSAR surface displacement signals and soil moisture conditions.

### 4.3  Validation

In order to quantify the validity of InSAR measurements for soil moisture retrieval, the in situ near-surface soil moisture dataset of Yamal was split into a calibration and validation dataset. For validation purposes, exclusively the data from Yamal

was selected due to its acquisition during typical moisture conditions, in contrast to the data from Inuvik, which was acquired during drought conditions. To address high heterogeneity of soil moisture patterns and to deal with differences in scale and geolocation of in situ compared to InSAR data, the in situ records were grouped into discrete bins representing 10 % volumetric soil moisture increments (comparable to Bartsch et al. (2020)). Median $\alpha_{DDT}$ values were then computed for each bin within the calibration dataset to establish a linear relationship. The coefficient of determination ($R^2$) was calculated to quantify the

strength of this relationship, while P-values were derived to ascertain its statistical significance. Additionally, the root mean square error (RMSE) was determined for the validation dataset to evaluate predictive accuracy. Additionally, classification accuracy was assessed for classifications comprising a total of 6 or 3 moisture level classes. To achieve this, the $\alpha_{DDT}$ values of samples from the validation dataset were converted into moisture values using the established linear relationship. Subsequently, these values were categorized into distinct moisture levels and compared to the true moisture classes.

### 4.4  Processing of auxiliary data

To assess the performance of $\alpha_{DDT}$ as a moisture indicator, other indices such as NDMI and TWI were also compared. The NDMI was calculated for the Inuvik and Yamal sites, where in situ near-surface soil moisture measurements were available, using Band 5 and 6 of the Level-2 (Bottom of Atmosphere) Landsat 8 data (Equation 2).

$$NDMI = (Band5 - Band6)/(Band5 + Band6) \tag{2}$$

The TWI is defined as

$$TWI = ln(SCA/tan\phi) \tag{3}$$

with SCA being the Specific Catchment Area and $\phi$ the slope angle. The ArcticDEM at 2 m spatial resolution was used for its calculation.

For the discussion of the results in the context of permafrost-specific landscape features, different land-cover units were

manually digitized based on UAV observations, high-resolution satellite images, and online services of Google and Esri Satellite maps. Areas of high- and low-centered polygons were differentiated from other tundra (with and without wetlands) for 25



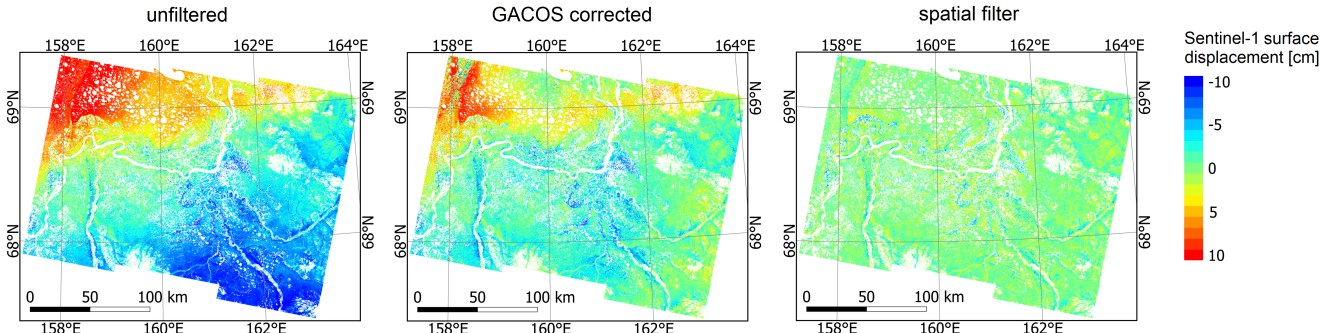

**Figure 4.** Comparison of unfiltered, GACOS corrected and spatial filter InSAR results for 23.06.2017 - 03.09.2017.

selected areas, each sized 1 km x 1 km. The average usage of different types of remote sensing data was 15% from UAV images, 77% from high-resolution satellite images, and 8% from online services. The average fraction of area covered by low-centered polygons is 3.9% (ranging from 0.4 to 11.2%). High-centered polygons cover a larger average fraction of about 12% (ranging

from 4.4 to 29.0%).

## 5 Results

### 5.1 Atmospheric correction

#### 5.1.1 Spatial filter radius for InSAR processing

In order to specify the filter radius, some initial tests were performed and visually evaluated. Supplement Figure S 2 depicts

results of a displacement map with no applied filter, a spatial filter of radius $\sim 60$ km and of radius 6 km. While atmospheric effects are still clearly noticable for the 60 km radius, they were mostly removed for the 6 km radius. Although it cannot be excluded that also large-scale deformation signals have been removed, a radius of about 6 km (512 pixels) was chosen for further investigations.

#### 5.1.2 Displacement time series, Chersky, Yamal and Inuvik

To investigate temporal InSAR subsidence results, we examined the displacement time series for selected sample points.

The points for Chersky were selected within an eastern and western part of the Sentinel-1 scene. They are located in areas of similar land cover within wet ecotopes, where summer subsidence can be expected (locations are shown in Figure 2). They represent different patterns of tropospheric delay (e.g. Supplement Figure S 1). While the differences at point A are not as pronounced for unfiltered and GACOS-corrected results, point B shows clear improvements with GACOS correction (Figure

5) . The implausible heave signal observed during the thawing seasons of the years 2017 and 2018 in the unfiltered results was mostly corrected when GACOS correction was applied. However, as illustrated in Figures 4 and 8a, this heave signal was not







**Figure 5.** Sample points displacements by degree day of thaw (DDT) for Chersky, accounting for early thaw data gap in InSAR time series by extrapolation (approach following Bartsch et al. (2019), dotted lines correspond to linearly extrapolation part of the time series). Point locations see Figure 2





corrected everywhere. This might be attributed to unfiltered atmospheric effects, or, as similar patterns were observed in two different years, also displacements at the reference point can not be ruled out. Unfiltered and GACOS-corrected results exhibit distinct temporal fluctuations, whereas spatially filtered results are clearly smoothed. However, this smoothing also results in a
reduction of overall magnitude of displacement values.

For the Yamal region, the displacement time series is investigated for a long-term in situ subsidence point on the CALM grid (Figure 6). While unfiltered and GACOS-corrected time series are very similar, spatially filtered data exhibit the same smoothing and reduction in subsidence magnitude as in the Chersky example. The depicted sample point, which exhibits an average subsidence of 3 cm, falls within the mean range of other points measured on the CALM grid, with a mean subsidence
of 4.7 cm $\pm$ 2 cm (standard deviation). The agreement with in situ subsidence values may vary with sample location, which is further investigated in the next section.

In situ subsidence values for the Inuvik region were available for different years than InSAR data but were nonetheless used to compare general magnitudes (Figure 7). In situ values were extrapolated similarly to InSAR data to account for the data gap between the first measurement and the start of the thawing season. The in situ subsidence rates of 2015 matched well with
InSAR data in terms of magnitudes. Higher rates, such as those observed in 2016, were also recorded with InSAR for the year 2023, particularly for point TVC1. However, for the spatially filtered results, the magnitude for 2023 at TVC1 was reduced. GACOS was able to correct for some of the fluctuations in the unfiltered results; however, in the case of 2019, it introduced additional artefacts.

### 5.1.3 In situ subsidence comparison, Yamal

The results of the InSAR time series were evaluated with in situ subsidence measurements at the CALM grid (Figure 2) on Yamal (Bartsch et al., 2019). Results for the years 2016, 2017, 2018 and 2021 were compared (Figure 9). For the year 2019 there was mostly no Sentinel-1 data available for this region and in 2020 missing acquisitions resulted in low coherence values preventing the generation of reliable results. The number of available in situ points was extended over the years and some points exhibit data gaps, however, for the 2D boxplots of in situ and InSAR values, all available data points were included,
along with an indication of the number of available samples (Figure 9). While unfiltered and GACOS-corrected plots show similar results, the spatially filtered results deviate and feature lower values, even heave, especially for the year 2016. All methods (unfiltered, GACOS-orrected, spatial filtering) exhibit lower annual mean InSAR subsidence signal values across all points with the longest available in situ time series compared to in situ measured values (Figure 9 bottom right). The lowest match to in situ values was derived for the spatially filtered results. The range of InSAR values is roughly similar for unfiltered
and GACOS-corrected results for these points, however while unfiltered values seem to disperse, GACOS-corrected values are less scattered.

### 5.1.4 Standard deviation of phase residuals for all regions

Focusing on the GACOS-corrected results, the standard deviations of the phase residuals are further investigated in comparison to unfiltered results (Figure 10). The phase residuals are derived through a comparison between simulated interferograms of a



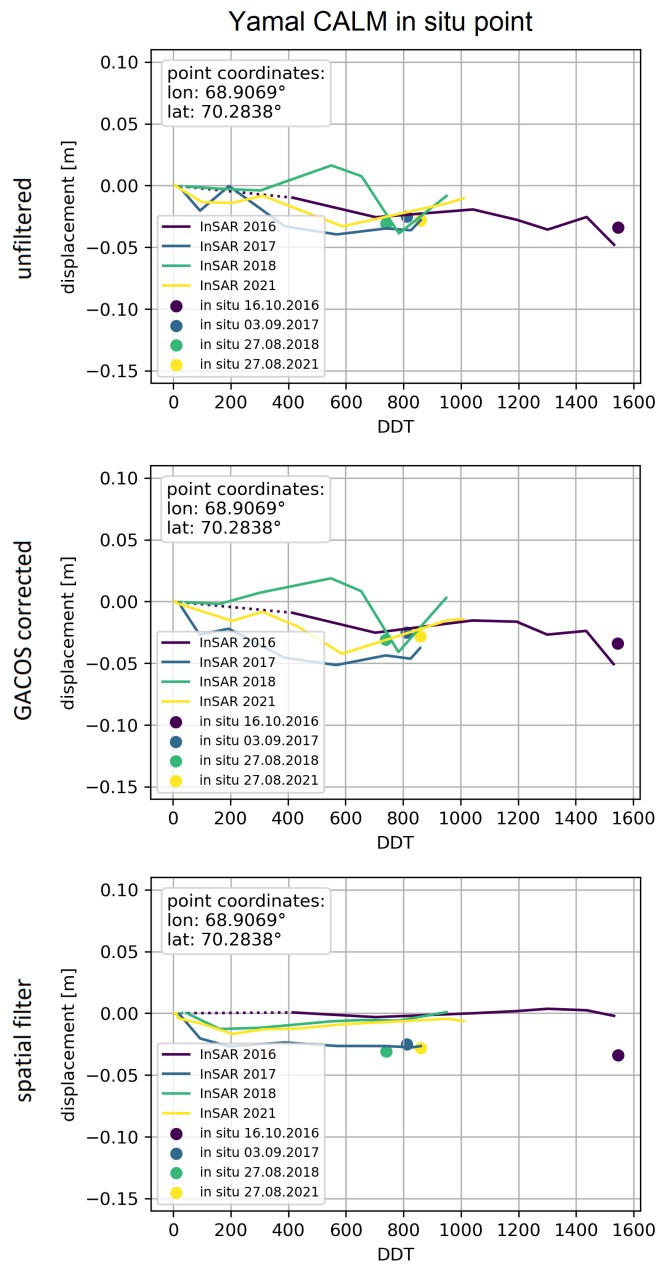

**Figure 6.** Displacements by degree day of thaw (DDT) of a sample point location with in situ subsidence measurements on the CALM grid on Yamal. The early thaw data gap in the InSAR time series was accounted for by extrapolation (approach following Bartsch et al. (2019), dotted lines correspond to linearly extrapolation part of the time series). CALM grid locations see Figure 2





**Figure 7.** Displacements by degree day of thaw (DDT) of two sample points with in situ subsidence measurements for the years 2015 and 2016 near the Trail Valley Creek (TVC) research station (Inuvik region). The early thaw data gap in the InSAR time series was accounted for by extrapolation (approach following Bartsch et al. (2019), dotted lines correspond to linearly extrapolation part of the time series). TVC location see Figure 2




(a) Chersky

(b) Inuvik

(c) Yamal

**Figure 8.** Subsidence rate $\alpha_{DDT}$ (median of all processed years) for (a) Chersky, (b) Inuvik, (c) Yamal. All regions are masked for slopes >5°. The Chersky region has additionally been masked for areas higher than 150 m due to temperature lower than 2°C or more (according to ERA5) than in lower regions during the Sentinel-1 acquisition time. (b) and (c) show subsets of the processed regions, depicting the area where in situ data is available.





**Figure 9.** 2D boxplots (depicting median values - filled circles; quartiles - boxes; minimum, maximum - whiskers; and outliers - empty circles) for in situ subsidence measured at the CALM grid on Yamal vs. InSAR subsidence for all available points per year (note available number of samples varying per year, indicated in box on the right hand side). Lower right: scatterplot of mean subsidence values for 6 long-term sample points (measured in 3 - 4 years) for each method and all available years (each point representing one year).

smoothed time series and the actual interferograms. Due to the absence of redundancy within the interferograms, the resulting values primarily reflect the effects of the time series smoothing. The statistics for the whole processed scenes for each thawing season were investigated, with terrain-slopes of $> 5°$ being masked out in order to exclude $\alpha_{DDT}$ values which may encounter erroneous results for vertical subsidence due to additional horizontal displacement resulting from by mass movements, such as solifluction. The values improve for the GACOS-corrected results (see Figure 10 and also Figure A3, depicting differences

between unfiltered and GACOS corrected results) especially for the Chersky and Inuvik region. However, for the Yamal region most years showed better values before the GACOS-correction.





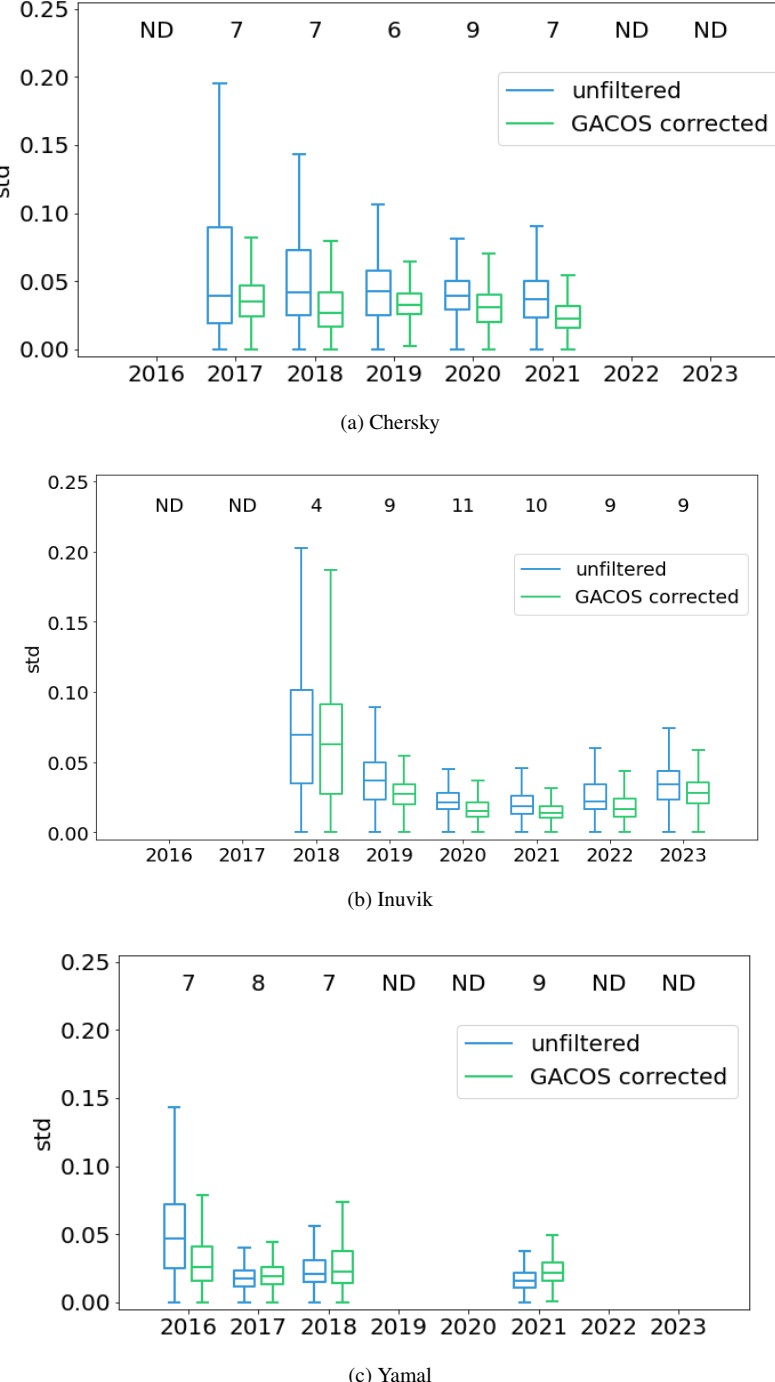

(a) Chersky

(b) Inuvik

(c) Yamal

**Figure 10.** Boxplots of standard deviations of the phase residuals for the whole processed scenes (terrain-slopes of >5° were masked out) for unfiltered and GACOS corrected results for (a) Chersky, (b) Inuvik and (c) Yamal. Number of used Sentinel-1 scenes specified on top of the plots.



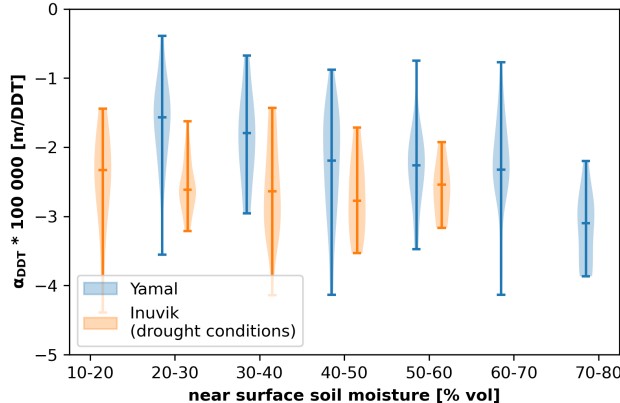

**Figure 11.** Distribution of GACOS corrected $\alpha_{DDT}$ values for near-surface soil moisture bins of Yamal and Inuvik (at minimum 6 samples per bin).

## 5.2 Soil moisture

### 5.2.1 Soil moisture comparison, Yamal and Inuvik

Comparing the displacement rates (displacement per DDT; $\alpha_{DDT}$ maps see Figure 8) to soil moisture measurements (Figure 11) showed higher $\alpha_{DDT}$ rates and therefore higher subsidence signal values for points with higher soil moisture for the Yamal study site. As the soil moisture measurements at the Inuvik site were conducted under drought conditions the soil moisture values tend to be lower. The relationship identified for the Yamal site was confirmed for moisture values below 50% vol. However, values for higher soil moisture deviated from this trend.

Similarly, the TWI and NDMI were compared to in situ near-surface soil moisture values. Higher TWI values were associated with higher soil moisture levels in the Yamal region (Figure 12). However, the comparison for the Inuvik region did not yield consistently increasing TWI values with rising soil moisture, with deviations found for moisture bins below 30% vol. The NDMI values exhibited only low spread, and their comparison with near-surface soil moisture data revealed no discernible stringent relationship. While NDMI values tend to be lower for low soil moisture levels on Yamal, the opposite is observable for the Inuvik region.

### 5.2.2 Accuracy assessment, Yamal

In order to derive a measure of quality for the applicability of $\alpha_{DDT}$ values as a soil moisture proxy, a linear relationship for the Yamal calibration dataset was derived (Figure 14). For this the median $\alpha_{DDT}$ values of 6 soil moisture bins were calculated. The number of sample points per soil moisture bin ranges from 4 (soil moisture range < 70 % vol) to 16 (soil moisture range 40 - 50 % vol). The standard deviations of $\alpha_{DDT}$ * 100 000 for each bin are below 0.97 (Table 4). The obtained linear regression has a coefficient of determination of $R^2 = 0.72$. While the intercept of the linear regression is statistically insignificant with a




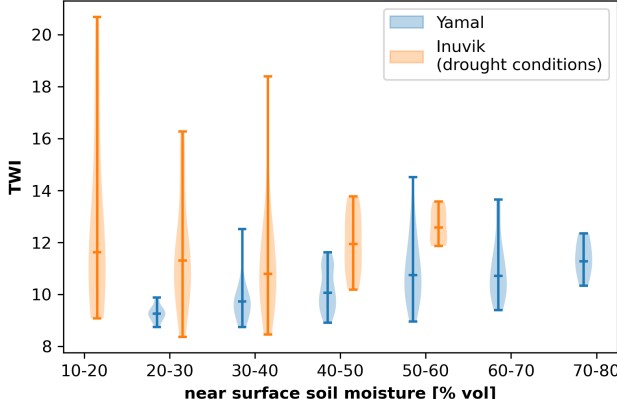

**Figure 12.** Distribution of TWI values for near-surface soil moisture bins of Yamal and Inuvik (at minimum 6 samples per bin).

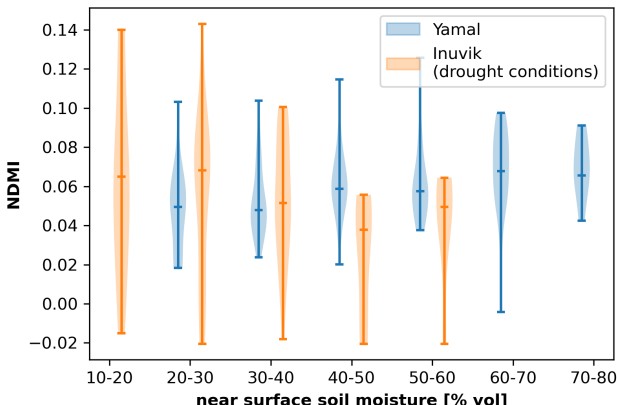

**Figure 13.** Distribution of NDMI values for near-surface soil moisture bins of Yamal and Inuvik (at minimum 6 samples per bin).

P-value of 0.91, the slope of the regression has a P-value of 0.03 indicating statistical significance. Values of $> 60 \%$ vol reveal to have greater deviations from what appears to be a nearly perfect correlation of values $< 60 \%$ vol, with P-values of 0.0036 and 0.0008 for intercept and slope, respectively.

Although the linear regression excluding values $> 60 \%$ appears to show a better fit, it does not follow physical intuition, as its intercept is -37.6 % vol and therefore not applicable for $\alpha_{DDT}$ values closer to 0. The derived equation for all 6 moisture bins (equation given in Figure 14) was subsequently used to predict soil moisture values of the validation dataset (Figure 15). A RMSE of 14 % vol was delineated for the validation data.

For the purpose of using $\alpha_{DDT}$ as an approximation for soil moisture conditions, we also calculated the accuracy of a possible classification of predicted soil moisture values within the soil moisture bins (indicated in Figure 14). For these 6






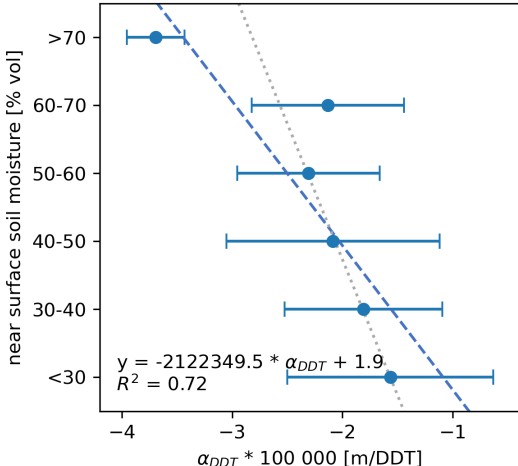

**Figure 14.** Relationship of averaged (10 % vol bins) in situ near-surface soil moisture of the Yamal calibration dataset with median $\alpha_{DDT}$ values for each bin. ($\alpha_{DDT}$ values represent median values of all years.) Error bars indicate standard deviation per bin. The dashed line depicts the linear regression of all depicted points (equation and $R^2$ indicated on bottom left). The dotted line shows the relationship of the moisture values < 60 % vol.

**Table 4.** Count and standard deviation of near surface soil moisture sample points (Yamal) for each bin value used in Figure 14

| soil moisture bin [% vol] | number of sample points | standard deviation of $\alpha_{DDT}$ * 100 000 [m/DDT] |
|---|---|---|
| <30 | 8 | 0.93 |
| 30-40 | 13 | 0.71 |
| 40-50 | 16 | 0.97 |
| 50-60 | 13 | 0.65 |
| 60-70 | 10 | 0.69 |
| >70 | 4 | 0.26 |

classes an accuracy of 25 % is achieved. A reduction of classes into 3 bins of < 40, 40-60 and > 60 % vol would result in an accuracy of 53 %.

To compare this accuracy assessment to other remotely sensed soil moisture products, we investigated values of ESA's CCI soil moisture product (Gruber et al., 2019; Dorigo et al., 2017; Preimesberger et al., 2021). Due to the coarser resolution of 0.25°, our in situ data is covered by only two pixels. The mean soil moisture values for the investigation period are only 19 - 22 % vol for the passive microwave product and 18 % vol for the combined solution (passive and active microwave, see Figure 16). These products have an RMSE of 30 % vol and 33 % vol for passive and combined solutions, respectively,



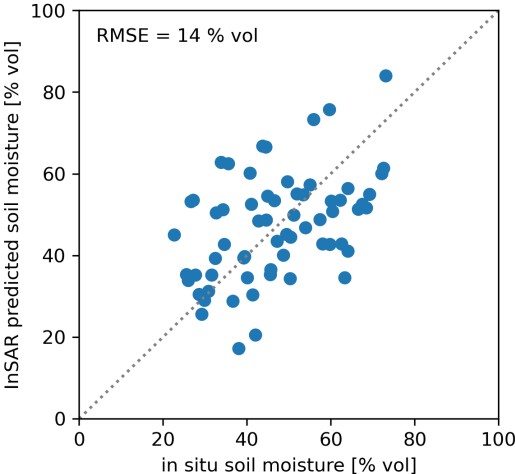

**Figure 15.** Scatterplot of in situ soil moisture values and calculated soil moisture values for the validation dataset of Yamal (used soil-moisture - $\alpha_{DDT}$ relationship see Figure 14). The 1:1 line is depicted as a grey dotted line.

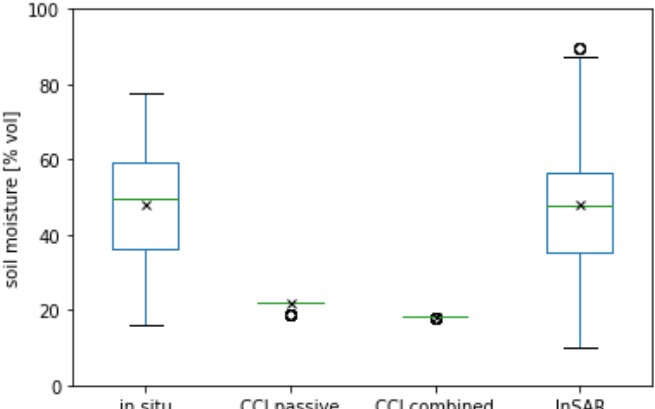

**Figure 16.** Boxplot of in situ measured near-surface soil moisture values compared to values of the CCI soil moisture products (passive, and combined product, (Gruber et al., 2019; Dorigo et al., 2017; Preimesberger et al., 2021)), and soil moisture values calculated with the InSAR approach for Yamal. Black x's indicate mean values.

# 6 Discussion

## 6.1 Soil moisture

Our InSAR derived soil moisture approach delivers static values intended to serve as a proxy for general soil moisture condi-

tions. While yearly products are feasible, median values are recommended to account for irregular values caused by uncorrected sources of error.



Our investigations showed higher subsidence values for points of higher in situ near-surface soil moisture compared to dryer ones (Figure 11). This effect could be demonstrated for both study sites, Yamal and Inuvik. The relationship was more pronounced at the Yamal site, where in situ data measurements where conducted under normal moisture conditions. This
observed relationship aligns with findings from Antonova et al. (2018), who identified more prominent subsidence in wetter parts of thermokarst basins. They attributed this to higher ground ice contents of these parts and higher ground heat flux found in wet parts due to their higher thermal conductivity.

Other wetness indices were compared with in situ near-surface soil moisture data, similar to previous $\alpha_{DDT}$ investigations. This analysis aimed to evaluate the performance of $\alpha_{DDT}$ relative to the other indices, specifically to determine whether these
indices also demonstrated strong correlations with the in situ measurements. The NDMI derived from multispectral data (Gao, 1996) exhibited varying correlations with soil moisture values across the investigated study sites at in situ sampling point locations (Figure 13). The TWI (Beven and Kirkby, 1979), a static wetness index derived from topographic information and often used as a proxy for soil moisture, exhibited similar performance to the InSAR results at the Yamal site (Figure 12). However, at the Inuvik site, the analysis did not indicate the suitability of TWI as a measure for near-surface soil moisture.
These findings suggest a better suitability of $\alpha_{DDT}$ for deriving near-surface soil moisture conditions.

To quantify the performance of the investigated method, results were evaluated for their correlation with in situ soil moisture measurements. An empirical function for a calibration dataset was derived and assessed for its accuracy using a validation dataset. The in situ dataset from Yamal was chosen as it was collected under typical moisture conditions rather than drought conditions. Binned and averaged in situ soil moisture and $\alpha_{DDT}$ data yielded a comparably high coefficient of determination
of 0.72 (Figure 14). It was noted that excluding higher soil moisture levels of > 60 % vol would result in nearly perfect correlation. However, to account for all relevant moisture levels, the function derived from all moisture values was used in the further course. This function also appears to be more plausible, as its intercept is 1.9 which would mean no deformation for basically dry soils. However, it should be noted that the coefficient of determination for unaveraged calibration data would be only 0.15. Binning was conducted (comparable to Bartsch et al. (2020)) to account for differences in the in situ data's
representativeness (point measurements versus spatial resolution of InSAR data). A comparison of in situ observed moisture levels to predicted ones for the validation dataset revealed no bigger discrepancies for higher soil moisture values (Figure 15). A RMSE of 14 % vol could be achieved.

Results were further compared to the CCI soil moisture product. A direct comparison between the CCI soil moisture dataset and the InSAR data is challenging due to the significant difference in their spatial resolutions. The CCI dataset has a much
coarser resolution, with each pixel covering an area approximately $10^6$ times larger than that covered by each pixel in the InSAR dataset. Consequently, any differences observed between the two datasets when comparing them to in situ data may be partly attributed to this resolution discrepancy. The CCI soil moisture comparison showed RSME values that are twice the value of the InSAR approach. The moisture values of the CCI products are much lower than was measured in the field (Figure 16), with just 1 % of field data being lower than what was indicated by the CCI soil moisture products (passive and passive- active
combined). The active product of CCI soil moisture was not used, as it provides only saturation values. However, comparisons of in situ values and the active product derived from ASCAT were previously performed by Högström et al. (2018) for five



Arctic regions. The in situ data were scaled to the satellite product to enable comparison. ASCAT has been shown to be drier than in situ in most cases, what agrees with our result.

We further assessed the accuracy of a potential soil moisture classification product based on $\alpha_{DDT}$ values. A classification
with 6 moisture classes would yield a 25 % accuracy. Therefore, it is suggested to only use 3 moisture categories, for which the accuracy would improve to 53 %. It should be noted that assessed soil moisture values used for the derivation of the linear relationship were within the range of 20 - 80 % vol. However, it can be assumed that a proposed classification of 3 moisture categories is conservative enough to also account for moisture values outside of the tested range.

## 6.2   DDT normalization

As shown in Figure 5, the time series of GACOS-corrected results still show interfering fluctuations. However, this issue may be addressed by simply utilizing the slope of the time series ($\alpha_{DDT}$). What is further visible (Figure 5) is the data-gap at the beginning of the time series, which may differ in length for different years or study regions. Nevertheless, this should be of lesser concern if a linear trend is to be assumed over DDT, which, based on our findings, appears to be mostly the case in our study regions. It is important to note, however, that active layer thaw is presumed to be grater in early summer and decreasing
in August and September (Short et al., 2014). Although this is not consistently reflected in our DDT plots (Figure 5), it cannot be completely ruled out.

The uncertainties for ERA5 data in the Arctic, which were used to derive DDT, represent another potential source of error. A comparison of ERA5 values with those from an ERA5 independent automatic weather station near Chersky revealed slightly lower measured values at the weather station than ERA5 values (see also Figure 19). The seasonal maxima of DDT for the
years 2016 - 2022 (missing data for 2018) differed by about 100 - 150℃, which corresponds to approximately 7 - 10% of the maximal ERA5 DDT values and would result in slightly different $\alpha_{DDT}$ values. DDTs of the automatic weather station yielded $\alpha_{DDT}$ values 17% lower than those of ERA5 (Figure 19).

It also needs to be considered that ERA5 2 m height air temperatures may deviate from top-soil temperatures, resulting in variations in $\alpha_{DDT}$ values. In order to investigate these variations, we compared ERA5 results to ground temperatures at 4
points on Yamal, measured between October 2016 and August 2017 near the surface using iButtons (Figure 2). While the point at the CALM grid showed good agreement with ERA5 values in the positive temperature range, the other points demonstrated lower in situ temperatures than ERA5 temperatures (Figure 17). Although the in situ time series ended on August 24th and maximum DDT values were not yet reached (80% of maximal ERA5 DDT value), we compared maxima of ERA5 and in situ DDT values for this date (aggregated from values shown in Figure 17). Only the in situ values of the CALM grid had higher
DDT values than ERA5 (147.2 higher, 21% of ERA5). The other points showed higher differences, ranging from 340 to 404, which is 48 to 55% of the maximal ERA5 value on August 24th. These differences can be explained by insulating vegetation cover, as ERA5 represents air temperature at 2 m compared to near-surface soil temperature. Furthermore, quality issues and coarse resolution of ERA5 data may play a role. The $\alpha_{DDT}$ values differ significantly from ERA5-derived $\alpha_{DDT}$ values in the range of -7 * $10^{-6}$ to 87 * $10^{-6}$ (Figure 18).



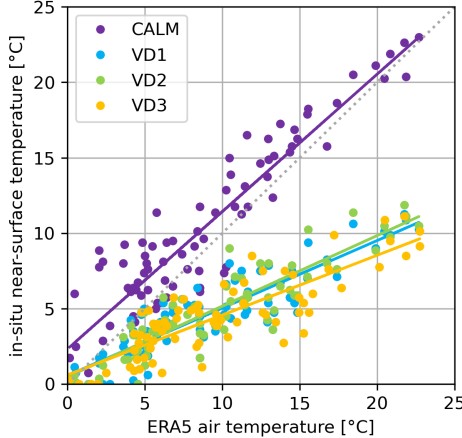

**Figure 17.** Comparison of in situ iButton near-surface soil temperatures (Table 2) and 2m height air temperatures derived from ERA5 for 4 different points on Yamal. In situ point locations see Figure 2. $R^2$ for CALM: 0.85, VD1: 0.79, VD2: 0.80, VD3: 0.71.

$\alpha_{DDT}$ derived with DDT values of even deeper soil layers would naturally deviate even more from those derived with ERA5 data (Figures 18 and 19). The resulting displacement rates ($\alpha_{DDT}$) are shown to be larger for deeper soil layers, as derived from borehole data of 50 cm depth (Yamal) and 4 cm and 8 cm depth (Chersky).

### 6.3 Atmospheric corrections

In order to achieve reliable InSAR results for soil moisture comparisons, the issue of atmospheric disturbances had to be
addressed. Investigations of displacement time series (Figures 5, 6 and 7) revealed that the applied spatial filter not only reduces spatial variations but also flattens temporal fluctuations. While a reduction in temporal variations seems to be more reasonable than the often high variability of unfiltered or GACOS-corrected results, the spatially filtered results mostly lacked any large scale spatial deformations, which may not always represent actual conditions. In this regard, a filter size of 6 km is too small; however, larger filter sizes would not account for small scale effects of the turbulent atmosphere. Comparisons with
in situ subsidence values (Figure 9) further illustrated that spatially filtered results were greatly reduced in magnitude. This not only led to lower subsidence values, but also resulted in measured heave signals, which deviate from in situ measured results.

GACOS-corrected products, on the other hand, showed promising results. After correcting GACOS data for encountered artefacts (see Supplement Figure S 1), the derived results were able to compensate for some improbable summer heave signals visible in the Chersky time series of unfiltered results (Figure 5). However, some atmospheric effects remained (Figure 4),
which may lead to the encountered fluctuations within the time series. It should also be noted that GACOS can sometimes introduce new artifacts (see also year 2019 of Figure 7) and may not necessarily guarantee an improvement of results due to scarcity of GPS stations and the coarse weather model resolution. However, the comparison of standard deviations of phase residuals showed improvements compared to unfiltered results, especially for the Chersky and the Inuvik region. As the differences of the median values (unfiltered - GACOS, Figure A3) seem to show a dependency on study area, this may indicate







**Figure 18.** InSAR Displacements of the thawing season 2017 by degree day of thaw (DDT) for 4 sample points on Yamal. DDT was derived from ERA5 2 m height air temperature, in situ near-surface iButton data and from borehole temperatures at 50 cm depth (time series shortened due to in situ data availability, see Table 2). Interpolations (linear regression) are plotted as dotted lines. In situ point locations see Figure 2.

varying effectiveness of GACOS corrections depending on region. The differing results for the Yamal region, however, also indicate that although in this area the GACOS corrections may not always lead to an improvement (see also Figure 6), some years may still benefit from this correction.

     Even though the standard deviation values of the phase residual were mostly slightly higher for GACOS-corrected results than for unfiltered ones in the Yamal region, the comparison to in situ subsidence data showed the closest match for GACOS 560    within this area.

     The overall lower InSAR subsidence compared to in situ values (Figure 9) has also been reported in other studies in Arctic regions (Short et al., 2014; Antonova et al., 2018) and may have multiple causes. First, in situ measurements are point measurements, and medium-scale InSAR measurements may therefore underestimate true displacements due to spatial aver-





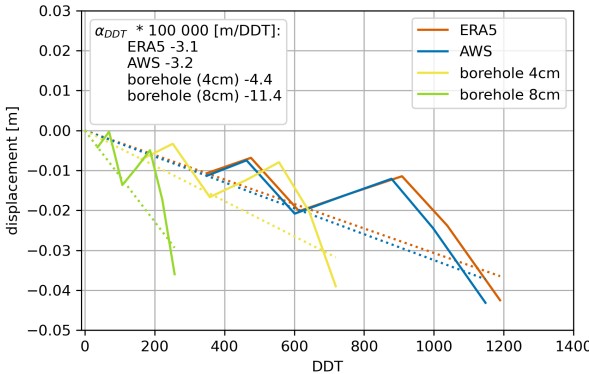

**Figure 19.** InSAR Displacements of the thawing season 2017 by degree day of thaw (DDT) for a sample point of the Chersky region. DDT was derived from ERA5 2 m height air temperature, an automatic weather station (AWS) data and from borehole temperatures at 4 cm and 8 cm depth. In situ point location see Figure 2.

aging (Short and Fraser, 2023). Furthermore, it cannot be excluded that the selected reference point (airstrip at Bovanenkovo)
also experiences some degree of surface deformation. The assumption of a linear relationship between DDT and thaw sub-
sidence may also play a role. Potential faster subsidence rates during the unmonitored beginning of the thawing season may
lead to an underestimation of subsidence values. In addition, soil moisture variations may have an effect on measured InSAR
results. Soil drying, which leads to a line-of sight shortening, results in an uplift signal (Zwieback et al., 2015), potentially
contributing to lower subsidence values compared to in situ measurements. The effect is estimated to be 10%-20% of the radar
wavelength (Zwieback et al., 2017). Further effects were reported due to vegetation growth and the inherent decrease in plant
moisture (Zwieback and Hajnsek, 2016), which may also cause additional uplift signals. An investigation of soil moisture
change at Trail Valley Creek (data source of soil moisture time series: Boike et al. (2023)) for consecutive Sentinel-1 acquisi-
tions, compared with changes in InSAR displacement signals (see Figure 20), revealed no significant relationship, leading to
the assumption that this source of error is of subordinate importance in this case. Furthermore, it should be noted that validat-
ing InSAR displacement values was not the primary objective of this study. An underestimation of subsidence values is less
relevant if it occurs consistently, as the linear relationship between displacement and moisture values was derived from these
biased data.

## 6.4 Limitations for InSAR-processing

The demonstrated relationship between soil moisture and InSAR subsidence signals exemplifies the potential of InSAR data to
derive maps of soil moisture classes needed for e.g. upscaling carbon fluxes and climate modeling. However, some inhibiting
factors have to be addressed. As illustrated, GACOS corrections are an adequate and easily implementable solution to reduce
atmospheric effects. Some remnants may subsist, especially effects of the turbulent atmosphere (Li et al., 2022). Furthermore,



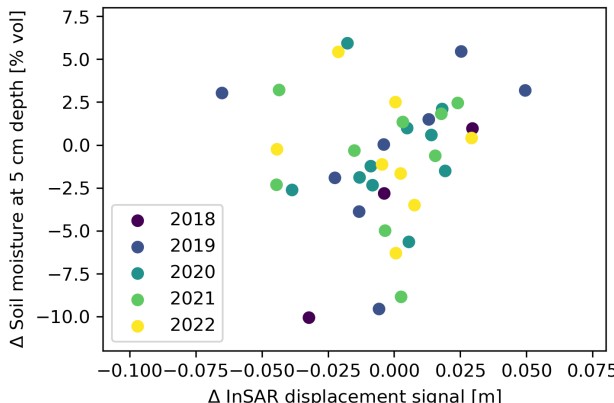

**Figure 20.** Comparison of InSAR displacement signals from consecutive Sentinel-1 acquisition dates and corresponding soil moisture change (derived from Boike et al. (2023)), measured at Trail Valley Creek in the Inuvik study area.

ionospheric effects are not accounted for with this correction. It is therefore essential to carefully select interferograms and use thawing seasons of years with minimal interfering effects in the derivation of soil moisture class maps.

Additionally, the selection of an adequate reference point is crucial and can prove to be difficult. Moreover, errors may increase with distance from the reference point (Short, 2017; Antonova et al., 2018), which has to be taken into account when processing whole Sentinel-1 scenes.

Another potential source of uncertainty is the impact of time-varying soil moisture on interferometric phase. A link between subsidence change and soil moisture change could not be observed at the study site of Trail Valley Creek (Figure 20). The

variation in soil moisture between consecutive acquisition dates is mostly below 10 % vol, which is much lower compared to other studies (Wig et al., 2023). Therefore the influence of soil moisture change appears subordinate compared to other disturbances (including atmosphere). However this phase term remains coupled with the ground motion information and thus persists as a potential source of error in the results.

## 6.5  Permafrost features characterization

Delineated soil wetness should also be reflected in certain land-cover features related to polygonal tundra. Slightly higher $\alpha_{DDT}$ and subsidence can indeed be observed for areas with polygonal features compared to other non-wetland tundra (Figures A1 and Figure A2). However, no visible differentiation is detectable between high-centered and low-centered polygons. The derived $\alpha_{DDT}$ values for the subsidence of the polygonal features are similar to those of wetlands without visible polygonal features. Both areas with low- and high-centered polygons are expected to be characterized by strong microtopography,

resulting in a mix of wet and dry parts, which leads to medium $\alpha_{DDT}$ values (not as low as for in situ points with very high soil moisture content, see Figure 11).





## 7 Conclusions

A representation of soil moisture classes is in high demand for applications in Arctic permafrost regions. In this study, we proposed a novel approach for deriving a soil moisture index based on InSAR subsidence signal rates. We illustrated the relationship between Sentinel-1 InSAR subsidence signal per DDT and surface soil moisture, with lower rates attributed to surface subsidence signals for drier regions and seasons, demonstrating its potential as a proxy for near-surface soil moisture classes. Compared to conventional coarser scale datasets such as ESA's CCI Soil Moisture product, which underestimated in situ values with an RMSE of around 30 % vol, our proposed method achieved an RMSE of 14 % vol. Although this approach provides only static information and does not account for seasonal fluctuations in soil moisture, it is assumed to be a valid indicator for general or predominant moisture conditions for permafrost regions. It is recommended to distinguish only three soil moisture categories. Its application for upscaling carbon fluxes and climate modeling remains to be tested.

Spatial patterns of wet and dry areas can be derived, but not all features typical for permafrost can be resolved. It is also pointed out that the ERA5 2 m height air temperature used for the calculation of DDTs may result in different subsidence rates ($\alpha_{DDT}$) compared to using top-soil temperatures.

Phase delays arising from soil moisture changes represent a limitation that should be considered in future studies, with potential benefits from leveraging insights gained from ongoing research.

Limiting factors for the utilisation of InSAR also include atmospheric effects. Atmospheric correction is therefore essential for InSAR applications to derive reliable results, especially in cases where coherence loss prevents the use of interferograms overlapping in time. We tested two easy to apply filtering methods for implementation in Arctic regions. Our study showed that while spatial filtering corrects for spatial and temporal variabilities, the suppression of larger scale displacement signals leads to a reduction of subsidence signal values, resulting in a poorer match with in situ values. GACOS-corrected results showed a reduction of atmospheric effects within the investigated time series, as well as an improvement in standard deviation values of phase residuals and the best match with in situ subsidence values. However, it should be noted that smaller scale tropospheric variations (<75 km) may not be captured (Murray et al., 2019). While long-wavelength components may be accounted for, the turbulent atmosphere phase mostly cannot be removed (Li et al., 2022). Nevertheless, our investigations showed that GACOS-corrected results are more suitable than spatial filtering and better suited for the derivation of soil moisture classes.

In upcoming years, InSAR processing of Sentinel-1 data is anticipated to benefit from the release of the Extended Timing Annotation Dataset (ETAD), which includes correction layers for tropospheric delay and ionospheric delay among others (Gisinger et al., 2022). Similar to GACOS the tropospheric delay is based on weather model data. However, data provision is currently only planned for newly acquired scenes, which prohibits its application for past years.

L-band missions, which have the potential to better preserve coherence, would reduce the importance of the investigated correction methods. Comprehensive L-band datasets are however mostly acquired on demand, and available data acquisition dates rarely meet requirements for the proposed investigation method. Nevertheless, with the launch of the upcoming NISAR mission, improved data coverage is anticipated in the near future.



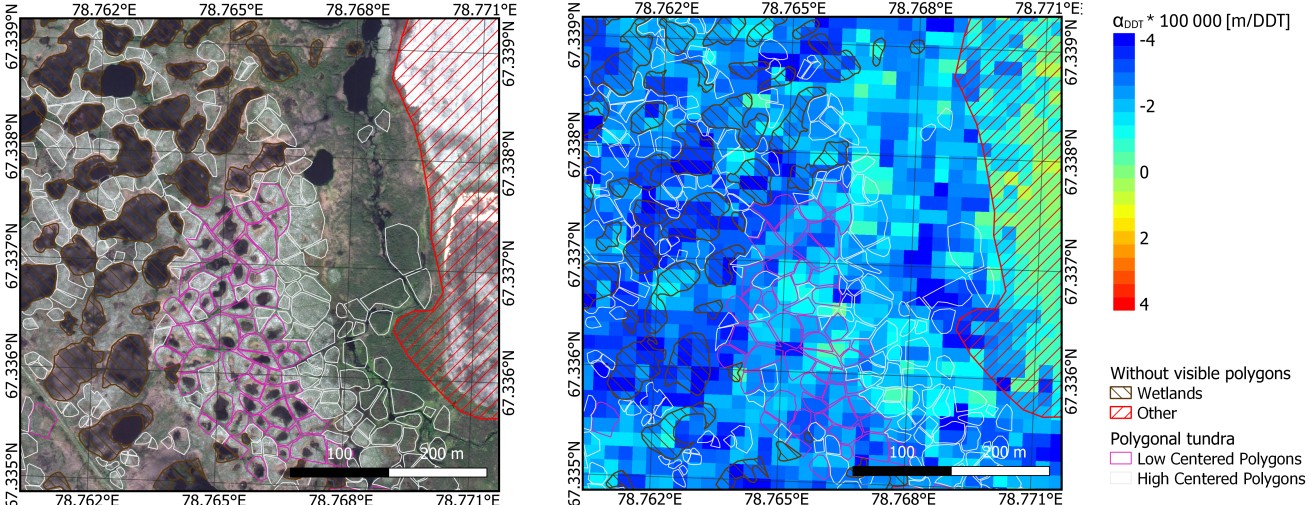

**Figure A1.** Subset of digitized land-cover features, left with WorldView-3 image as background map and right with $\alpha_{DDT}$.

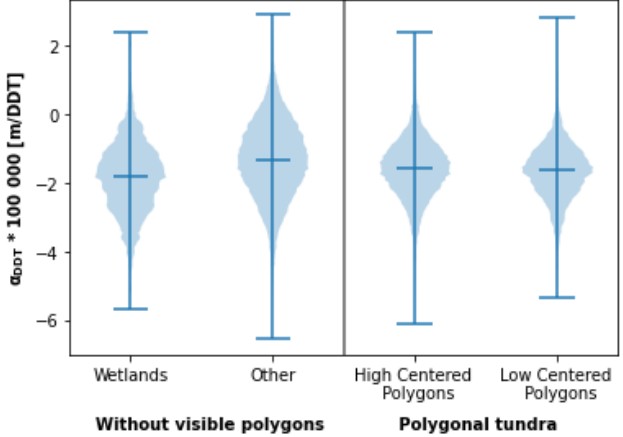

**Figure A2.** $\alpha_{DDT}$ values for different land-cover features at the Tazovsky study region.

## 8  Appendix

*Author contributions.* BW and AB developed the concept for the study. BW analysed the results and wrote the first draft of the manuscript. TS and NJ provided expertise on InSAR processing. AK, EB, ML, RK , HS, CB, XM and MG contributed to the in situ surveys and evaluation data processing. AB, TS, ML, RK and MG contributed to the writing of the manuscript.





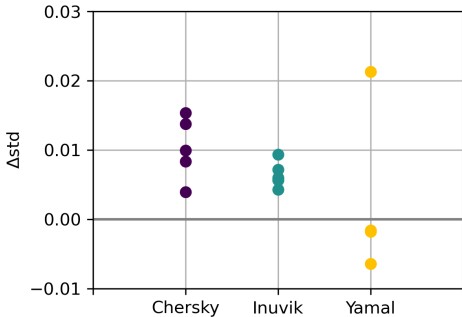

**Figure A3.** Difference of the in Figure 10 depicted median values of unfiltered - GACOS corrected values for each year, differentiated by study site.

*Competing interests.* The authors declare no competing interests.

640 *Acknowledgements.* We would like to thank Christophe Magnard of GAMMA Remote Sensing for providing the implementation of the GACOS correction within the Gamma software and Veronika Doepper, Helmholtz Einstein International Berlin Research School in Data Science (HEIBRIDS) for field work support in the framework of the AWI CA-Land_NWCanada2023 expedition.

Yamal field data was obtained within the framework of the state assignment from the Ministry of Science and Higher Education RF (theme No. FWRZ-2021-0012).

645 This work was supported by the European Research Council project No. 951288 (Q-Arctic) and the European Space Agency project CCI+ Permafrost (4000123681/18/I-NB).



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
