# Peer review of "InSAR-derived seasonal subsidence rates reflect spatial soil moisture patterns in Arctic lowland permafrost regions"

_EGUsphere, 2024_

## Referee Comment (RC2)

Widhalm and co-authors introduce a new soil moisture index tailored for Arctic lowland regions by using the ratio of seasonal thaw subsidence and thawing degree days (they called it as 'subsidence rate'). Utilizing InSAR subsidence observations from three Arctic sites, the study demonstrates a stronger correlation between this new index and in-situ soil moisture compared to traditional indices such as NDWI and TWI. The authors also explore the impact of methodological choices, such as InSAR tropospheric delay corrections and using 2 m air temperature vs. ground surface temperature in thaw index calculation, on the InSAR time series and its relationship with thawing degree days.

This research marks a new advancement in the application of InSAR for Arctic studies. However, there are several areas where the manuscript could be further refined. And I hope my comments can help improve the rigor, clarity, and focus of this very nice piece of work.

0. Terminology: "Seasonal Subsidence Rates"

This is a minor comment. The term "seasonal subsidence rates" used throughout the manuscript, including in the title, could be misleading because it implies changing rates over time. Instead, the proposed index represents a ratio of subsidence to thaw degree days, or subsidence normalized by degree days. If the authors choose to retain this 'rate' term, clarification should be provided at its first mention. Alternatively, a more precise term could be coined for this new index.

1. Soil Moisture Estimates

1a. The key results of this work should be about soil moisture and the new index (alpha). Fig. 8 does present maps of alpha over three regions. But how could readers interpret them? There's also a counter-intuitive representation of negative alpha values (see my comment #5).

1b. Volumetric soil moisture like those presented in Section 5.2 would be more useful output. But what is missing are maps of (categorized) soil moisture derived from the InSAR-based alpha. Is it possible for the authors to include them, which would greatly enhance this work's utility?

1c. It is worth adding further elaboration and discussion on the depth of soil moisture this new index reflects. Terms like 'near-surface' (line 9 and numerous places), 'general' (lines 464, 610,), 'top' (line 614) all imply a shallow depth. However, considering that thaw subsidence measured from InSAR essentially integrates responses from the entire thawed soil column (Liu et al., 2012; Chen et al., 2023), it seems likely that alpha reflects a weighted average of soil moisture within the thawed active layer. Because soil moisture and ice content in Arctic lowlands have strong vertical variations, it would be necessary to make clarification on the depth sensitivity. This also

helps when comparing alpha with other soil moisture products and indices such as ESA CCI (passive and combined) and NDMI.

1d. If possible, please specify the depth of in situ soil moisture measurements in Table 2, as this information is crucial for interpreting the results.

**2. Normalizing with Thawing Degree Days (DDT)**

This work proposes to scale seasonal thaw subsidence with DDT. Below, I lay out a theoretic framework based on Stefan's equation to give an alternative scaling scheme with the square root of DDT.

One form of Stefan equation for time-varying thaw depth $D(t)$ is (e.g., Kurylyk and Hayashi, 2015)

$$D(t) = \sqrt{\frac{2k \cdot DDT(t)}{\phi \rho_w L}}$$

where $k$ is the bulk thermal conductivity of the upper thawed soil, $L$ is the latent heat, $\phi$ is the volumetric moisture content, and $\rho_w$ is the water density.

To the first order, the magnitude of seasonal thaw subsidence is proportional to thaw depth times volumetric soil moisture ($D * \phi$), therefore

$$\text{Thaw Subsidence} \propto \sqrt{DDT \, \phi}$$

This $\sqrt{DDT}$ dependency serves as the basis for several previous studies (e.g., Liu et al. 2012; Hu et al., 2018) and can capture faster subsidence at the beginning of thaw season (line 335).

It is up to the authors, but it should be very straightforward if they decide to test this alternative scaling scheme. And if it turns out that square-root-of DDT works better, the theoretic framework can be easily refined to build a strong physics base for soil moisture retrieval.

**4. Tropospheric Delay Correction**

I agree with the authors that it is important to correct atmospheric (tropospheric plus ionospheric) phase delay in interferograms. The manuscript presents a valuable comparison of uncorrected, spatially filtered, and GACOS-corrected InSAR results, and points that GACOS is helpful in some cases but not in all cases. Such a comparison is informative and insightful. However, given the complexity and importance of tropospheric delay correction in InSAR studies on Arctic permafrost, my concern is evaluating the effectiveness and accuracy of the tropospheric delay

correction methods deserves a separate study by itself and may not suit the interest of TC readership.

For instance, the assessment presented in this manuscript is largely based on visual inspection (e.g., Fig 4, Fig 6, Fig 9) but lacks quantitative analysis. The spatial filtering is a simplified version of spatial-temporal filtering that is commonly used in InSAR time series analysis. Ideally, spatial-temporal filtering should be included in the comparison. And there are exemplary studies comparing various correction methods (e.g., Bekaert et al., 2015; Murray et al., 2019), none has been done for Arctic permafrost studies.

A more comprehensive and thorough evaluation is outside the scope of the current study and is better suited for a separate publication.

One way to sharpen the focus of this manuscript on the new soil moisture index is to emphasize the importance of atmospheric correction and to put visual comparisons into supplementary materials. This also helps to shorten the lengthy manuscript in its current form.

5. The manuscript does not explicitly state whether InSAR line-of-sight deformation has been converted to vertical displacement (or not). Clarification on this point is needed. Additionally, the manuscript adopts a convention to use negative values for subsidence (which is fine), but leaves the new index (alpha) to be negative. It is confusing as a more negative alpha means higher soil moisture. It should be more intuitive to reverse the sign in the definition of alpha (eq. 1) so that a higher positive alpha means higher soil moisture. Reversing the sign in the definition could enhance its interpretability and align conceptually with other soil moisture indices.

References:

Bekaert, D. P. S., Walters, R. J., Wright, T. J., Hooper, A. J., & Parker, D. J. (2015). Statistical comparison of InSAR tropospheric correction techniques. *Remote Sensing of Environment*, *170*, 40-47.

Hu, Y., Liu, L., Larson, K.M., Schaefer, K.M., Zhang, J., & Yao, Y. (2018). GPS Interferometric Reflectometry reveals cyclic elevation changes in thaw and freezing seasons in a permafrost area (Barrow, Alaska). *Geophysical Research Letters*, 45, 5581–5589.

Kurylyk, B. L., and Hayashi, M. (2016). Improved Stefan Equation Correction Factors to Accommodate Sensible Heat Storage during Soil Freezing or Thawing. *Permafrost and Periglac. Process.*, 27: 189–203.

Liu, L., Schaefer, K., Zhang, T., & Wahr, J. (2012). Estimating 1992–2000 average active layer thickness on the Alaskan North Slope from remotely sensed surface subsidence. *Journal of Geophysical Research: Earth Surface*, 117, F01005.

Murray, K. D., Bekaert, D. P., & Lohman, R. B. (2019). Tropospheric corrections for InSAR: Statistical assessments and applications to the Central United States and Mexico. *Remote Sensing of Environment*, *232*, 111326.

Wig's preprint has been published as this journal paper:

Wig, E., Michaelides, R., and Zebker, H. (2024). Fine-Resolution Measurement of Soil Moisture from Cumulative InSAR Closure Phase. *IEEE Transactions on Geoscience and Remote Sensing*, 62, 5212315.

---

## Author Comment (AC2)

[Figure]

*Figure 1: Comparison of displacement time series for two sample points with in situ subsidence measurements for the years 2015 and 2016 near the Trail Valley Creek (TVC) research station (Inuvik region). The top figures display GACOS corrected results in the DDT domain while the lower figures show GACOS corrected results in the $\sqrt{DDT}$ domain.*

---

## Author Response (AR1)

RC1:

1)

Line 87: The authors state that "Seasonally aggregated vertical displacement is in the order of a few centimeters". But this is not strictly true, this is just what is typically detected by medium resolution InSAR systems. True seasonal displacements are more commonly on the order of <10 cm, they can easily be <15 cm, or I have even heard of <20 cm from field measurements in very dynamic regions. This same statement is also in the abstract (line 6: "exhibit magnitudes of several centimeters"). It would be good to correct this.

> Reply: We made the following corrections:
>
> Abstract:
>
> - old: Seasonal thawing and freezing of nearsurface soil lead to subsidence-heave cycles in the presence of ground ice, which can exhibit magnitudes of several centimeters.
> - New: ... which exhibit magnitudes of typically less than 10 cm.
>
> Line 87:
>
> - Seasonally aggregated vertical displacement detected by InSAR is in the order of a several centimeters (e.g. Strozzi et al. (2018)). The magnitude can vary from year to year depending on the warming of the soil or changes in water content through variations in the water budget and is typically less than 10 cm but can exceed this in more dynamic regions.

2)

I have limited experience with the GACOS products. If the products are modelled at 6 hour intervals, is it possible that some sites have SAR acquisition times that line up more closely with the GACOS model results and therefore deliver generally better results? Six hours is quite a long time for the atmospheric patterns to change, it is not surprising to me that they may or may not help.

&

425-6. Is there any reason you can think of why the GACOS corrections are more effective at the Chersky and Inuvik sites than at Yamal? It comes back to my earlier question of how acquisition time of day intersects with GACOS modelling.

> Reply: We added the following sentences to section 6.3 in order to address these comments:
>
> - One reason for the performance differences observed in various regions may be the coarse temporal resolution of the weather model used in GACOS for the turbulent component. Although corrections are provided for the specific times of satellite acquisitions, the interpolated solution may align more closely with the 6-hour intervals of the weather model in some areas than in others. Moreover, the limited availability of GPS stations in certain regions may also contribute to these variations.

3)

The alphaDDT results are always presented as alphaDDT *100,000 (m/DDT). I did get used to it, but found it very unintuitive. Would the authors consider using mm/DDT instead? The results would have to be displayed with a couple of decimal places, but at least they would be more immediately meaningful.

Reply: We agree to the suggestion and adapted figures, table and text accordingly.

4)

Technical corrections

Reply: All suggested corrections were implemented accordingly

RC2:

0. Terminology: "Seasonal Subsidence Rates" This is a minor comment. The term "seasonal subsidence rates" used throughout the manuscript, including in the title, could be misleading because it implies changing rates over time. Instead, the proposed index represents a ratio of subsidence to thaw degree days, or subsidence normalized by degree days. If the authors choose to retain this 'rate' term, clarification should be provided at its first mention. Alternatively, a more precise term could be coined for this new index.

> Reply: We agree with the suggestion to clarify the term upon its first mention.
>
> We added the following in the abstract:
>
> … subsidence rates (calculated per thawing degree days - a measure of seasonal heating)
>
> And in the introduction:
>
> Hereafter, all references to subsidence rates refer to the DDT domain.

1. Soil Moisture Estimates

1a. The key results of this work should be about soil moisture and the new index (alpha). Fig. 8 does present maps of alpha over three regions. But how could readers interpret them? There's also a counter-intuitive representation of negative alpha values (see my comment #5).

&

1b. Volumetric soil moisture like those presented in Section 5.2 would be more useful output. But what is missing are maps of (categorized) soil moisture derived from the InSAR-based alpha. Is it possible for the authors to include them, which would greatly enhance this work's utility?

> Reply: We exchanged the Figure presenting maps of alpha with maps of categorized soil moisture derived from the InSAR-based alpha.

1c. It is worth adding further elaboration and discussion on the depth of soil moisture this new index reflects. Terms like 'near-surface' (line 9 and numerous places), 'general' (lines 464, 610,), 'top' (line 614) all imply a shallow depth. However, considering that thaw subsidence measured from InSAR essentially integrates responses from the entire thawed soil column (Liu et al., 2012; Chen et al., 2023), it seems likely that alpha reflects a weighted average of soil moisture within the thawed active layer. Because soil moisture and ice content in Arctic lowlands have strong vertical variations, it would be necessary to make clarification on the depth sensitivity. This also helps when comparing alpha with other soil moisture products and indices such as ESA CCI (passive and combined) and NDMI.

> Reply: We agree that the term "near-surface" should be revised, and did so where appropriate. We further elaborated the topic in the discussion:
>
> It is important to note that the relationship between $\alpha$DDT and soil moisture was derived using in situ measurements of near-surface soil moisture. However, since thaw subsidence observed via InSAR reflects an integrated response from the entire thawed soil column (Liu et al., 2012; Chen et al., 2023) $\alpha$DDT likely represents a weighted

average of soil moisture across the active layer. Given the pronounced vertical variations in soil moisture and ice content in Arctic lowlands, using in situ near-surface soil moisture data may introduce potential uncertainty when interpreting InSAR-derived soil moisture as representative of the entire active layer.

1d. If possible, please specify the depth of in situ soil moisture measurements in Table 2, as this information is crucial for interpreting the results.

> Reply: The soil moisture is measured at the top 5 cm. We added this information to Table 2.

2. Normalizing with Thawing Degree Days (DDT) This work proposes to scale seasonal thaw subsidence with DDT. Below, I lay out a theoretic framework based on Stefan's equation to give an alternative scaling scheme with the square root of DDT. One form of Stefan equation for time-varying thaw depth $D(t)$ is (e.g., Kurylyk and Hayashi, 2015) …

where $k$ is the bulk thermal conductivity of the upper thawed soil, $L$ is the latent heat, $\Phi$ is the volumetric moisture content, and $\rho_w$ is the water density. To the first order, the magnitude of seasonal thaw subsidence is proportional to thaw depth times volumetric soil moisture ($(D * \Phi)$), therefore …

This $\sqrt{DDT}$ dependency serves as the basis for several previous studies (e.g., Liu et al. 2012; Hu et al., 2018) and can capture faster subsidence at the beginning of thaw season (line 335). It is up to the authors, but it should be very straightforward if they decide to test this alternative scaling scheme. And if it turns out that square-root-of DDT works better, the theoretic framework can be easily refined to build a strong physics base for soil moisture retrieval.

> Reply: We tested this alternative scaling scheme, focusing on the preferred GACOS-corrected results for Inuvik. Comparisons with in-situ subsidence values indicate a better fit when using the original DDT domain especially for point TVC2. We added the according figure to the supplements. We also tested the influence of the square root of DDT on the delineation of the derived soil moisture relationship and whether this would improve accuracies. We have expanded figure 8 to incorporate the new results. R2 is reduced from 0.72 to 0.68. We added the following text:
>
> 4 Methods:
>
> Alternatively, a dependency on $\sqrt{DDT}$ has been employed in several previous studies (e.g. Liu et al. (2012); Hu et al. (2018))
>
> 4.2 DDT and subsidence relationship:
>
> Therefore, as an alternative approach, $\alpha\sqrt{DDT}$, was tested, incorporating a dependency on $\sqrt{DDT}$ (Liu et al., 2012; Hu et al., 2018; Liu, 2024)
>
> 4.3 Validation:
>
> Median $\alpha DDT$ values were then computed for each bin within the calibration dataset to establish a linear relationship and compared to the results derived using $\alpha\sqrt{DDT}$ from the alternative approach.

**5.1.2 Displacement time series, Chersky, Yamal and Inuvik**

Comparison of GACOS results in the $\sqrt{DDT}$ domain (Supplement Figure S3) shows that the in situ data align more closely with the results in the DDT domain.

**5.2.2 Accuracy assessment, Yamal**

The obtained linear regression has a coefficient of determination of $R^2 = 0.72$ for the $\alpha DDT$ values (Figure 7a) compared to 0.68 of the $\alpha\sqrt{DDT}$ values (Figure 7b).

**6.1 Soil moisture**

Binned and averaged in situ soil moisture and $\alpha DDT$ data yielded a comparably high coefficient of determination of 0.72 (Figure 7a), slightly higher than the alternative approach based on $\sqrt{DDT}$ values (Figure 7b)

**4. Tropospheric Delay Correction**

I agree with the authors that it is important to correct atmospheric (tropospheric plus ionospheric) phase delay in interferograms. The manuscript presents a valuable comparison of uncorrected, spatially filtered, and GACOS-corrected InSAR results, and points that GACOS is helpful in some cases but not in all cases. Such a comparison is informative and insightful. However, given the complexity and importance of tropospheric delay correction in InSAR studies on Arctic permafrost, my concern is evaluating the effectiveness and accuracy of the tropospheric delay correction methods deserves a separate study by itself and may not suit the interest of TC readership.

For instance, the assessment presented in this manuscript is largely based on visual inspection (e.g., Fig 4, Fig 6, Fig 9) but lacks quantitative analysis. The spatial filtering is a simplified version of spatial-temporal filtering that is commonly used in InSAR time series analysis. Ideally, spatial-temporal filtering should be included in the comparison. And there are exemplary studies comparing various correction methods (e.g., Bekaert et al., 2015; Murray et al., 2019), none has been done for Arctic permafrost studies.

A more comprehensive and thorough evaluation is outside the scope of the current study and is better suited for a separate publication.

> Reply: We agree to focus the manuscript on soil moisture and moved visual comparisons of Figures 4-7 to the appendix (A1-A4). However, we did not move Figure 9 (new index 4), as we consider this one of our key results. Regarding spatial-temporal filtering, we acknowledge its importance in InSAR time series analysis. However, since alpha represents the linear regression rate, additional temporal filtering may not significantly impact results for this parameter. We added the following text to section 6.3 Atmospheric corrections:
>
> While additional temporal filtering is recognized as an important aspect of InSAR time series analysis, it was not tested in this study, as $\alpha DDT$ , representing the linear regression rate, is unlikely to be significantly influenced by further temporal filtering.

5. The manuscript does not explicitly state whether InSAR line-of-sight deformation has been converted to vertical displacement (or not). Clarification on this point is needed. Additionally,

the manuscript adopts a convention to use negative values for subsidence (which is fine), but leaves the new index (alpha) to be negative. It is confusing as a more negative alpha means higher soil moisture. It should be more intuitive to reverse the sign in the definition of alpha (eq. 1) so that a higher positive alpha means higher soil moisture. Reversing the sign in the definition could enhance its interpretability and align conceptually with other soil moisture indices.

Reply: In line 305 of the InSAR processing section, the use of vertical displacements is already mentioned. Regarding the sign of the index, we reversed the sign of alpha for greater clarity and intuitiveness, and adjusted all affected figures and the equation accordingly. We added the following text:

... (Equation 1), reversing the sign in the equation to produce higher positive values for areas with higher soil moisture.

---

## Author Response (AR2)

Dear editor,

We now adapted the title to:

InSAR-derived seasonal subsidence reflects spatial soil moisture patterns in Arctic lowland permafrost regions

Sincerely,

Barbara Widhalm